# Operational Modal Analysis and Non-Linear Dynamic Simulations of a Prototype Low-Rise Masonry Building

Ilaria Capanna, Riccardo Cirella [ID], Angelo Aloisio *[ID], Franco Di Fabio [ID] and Massimo Fragiacomo [ID]

Civil and Environmental Engineering Department, University of L'Aquila, 67100 L'Aquila, Italy; ilaria.capanna@graduate.univaq.it (I.C.); riccardo.cirella@graduate.univaq.it (R.C.); franco.difabio@univaq.it (F.D.F.); massimo.fragiacomo@univaq.it (M.F.)
* Correspondence: angelo.aloisio1@univaq.it

**Abstract:** This paper focuses on the dynamic behaviour of a low-rise masonry building representing the Italian residential heritage through experimental and numerical analyses. The authors discuss an application of combined Operational Modal Analysis and Finite Element Model updating for indirect estimation of the structural parameters. Two ambient vibration tests were carried out to estimate the structure's dynamic behaviour in operational conditions. The first experimental setup consisted of accelerometers gathered in a row along the first floor to characterize the local dynamic of the floor. Conversely, the second setup had the accelerometers placed at the building's corners to characterize the global dynamics. The outcomes of the first setup were used to estimate the mechanical parameters of the floor, while the ones form the second were used to characterize the mechanical parameters of the masonry piers. Therefore, two finite element models were implemented: (i) a single beam with an equivalent section of the floor to grasp the local behaviour of the investigated horizontal structure; (ii) an equivalent frame model of the entire building to characterise the global dynamic behaviour. The model updating process was developed in two phases to seize local and global dynamic responses. The updated numerical model formed the basis for a sensitivity analysis using the modelling parameters. The authors chose to delve into the influence of the floor on the dynamic behaviour of low-rise masonry buildings. With this aim, non-linear dynamic analyses were carried out under different mechanical characteristics of floors, expressing the scatter for ordinary masonry buildings. The displacements' trends along the height of the building evidenced the notable role of the floor's stiffness in the non-linear dynamic behaviour of the building. Lastly, the authors derived the fragility curves predicting the seismic performance in failure probability under a highly severe damage state.

**Keywords:** ordinary masonry buildings; dynamic identification; non-linear dynamic analysis; fragility curves

## 1. Introduction

The evaluation of the dynamic behaviour of existing masonry buildings has received increasing attention in recent decades due to their unsatisfactory performances showed under seismic actions [1–4]. A significant proportion of the scientific literature about the dynamic behaviour of masonry buildings focuses on investigation of mechanical characteristics of vertical structures and horizontal ones, through experimental and numerical analysis, to grasp the main vulnerabilities of the built heritage. Despite the capability of experimental investigations to achieve in-depth knowledge of masonry buildings, few testing campaigns are carried out on full-scale specimens [5–9]. Within the experimental analysis, Operational Modal Analysis (OMA) directly evaluates the structural dynamic behaviour, allowing an accurate validation of Finite Element models (FE) [10,11] or structural models obtained using the applied element method (AEM) [12,13]. Nevertheless, the main difficulties in estimating the dynamic response of masonry structures through OMA stand in

the low-amplitude vibration in operational conditions and the presence of localized mode shapes. The low-amplitude vibration, mainly related to the massive nature of masonry constructions, entails the necessity of adopting sensors of adequate performance in terms of sensitivity and signal-to-noise ratio. Localized mode shapes require installing an adequate number of sensors to achieve a fine discretization of the experimental mode shapes. These difficulties have determined more extensive applications of OMA in reinforced concrete (RC) and steel buildings compared to masonry ones. Therefore, the application of OMA for the identification of the modal parameters of masonry structures as a practice-oriented non-destructive diagnosis technique is limited.

Still, dynamic identification using an adequate number of sensors with high-level performance may provide valuable information about the mechanical properties of masonry and the effect of retrofitting interventions. Masonry structures manifest a significant scatter of their mechanical parameters.

Non Destructive Tests (NDT) or semi-NDT are necessary for masonry structures due to the intrinsic heterogeneity of their features and consequent variety of their mechanical properties.

Semi-NDT give insight into the mechanical properties of a limited area where the tests are executed.

Still, the mechanical properties of a limited area are more valuable than pure estimation based on visual inspections and classification approaches.

However, dynamic tests may deliver comprehensive knowledge into the homogenized mechanical parameters by updating a finite element model using experimental modal parameters, such as mode shapes, modal frequencies and damping ratio. The model updating of a masonry structure using the modal parameters estimated from OMA can return an indirect estimation of the Young's modulus, weight and other structural parameters useful for addressing and refining structural investigations. There are a few applications of combined OMA and model updating in masonry structures due to the high uncertainties related to estimating the modal parameters of low-rise masonry structures from OMA and the difficulty in matching the measured response with the one simulated by the conventional, simplified structural models adopted for masonry structures.

Within the framework of numerical analysis, non-linear analysis is the most accurate approach to investigate the dynamic behaviour of masonry buildings, although reliable models for calculating the inelastic response to seismic input of load-bearing masonry structures are required [14,15]

Still, the complexity of the seismic behaviour of masonry buildings is sometimes exasperated by the sensitivity of the numerical tools used to the input variables [16–19]

Due to the reasons above, the design and seismic assessment of traditional masonry buildings are sometimes conventional, lacking a solid predictive model of the structural response. Therefore, understanding their dynamic behaviour represents a crucial issue in research for a possible transition between conventional design approaches to ones with a more appropriate mechanical base.

In this research, the authors present an investigation of the dynamic behaviour of a two-storey masonry structure, chosen as representative of the traditional low-rise residential masonry buildings widespread central–southern Italy, through in-depth experimental and numerical analyses.

Firstly, OMA are carried out. The modal parameters of the buildings are investigated using two configurations to obtain a better estimation for both the global and localized mode shapes. In fact, the presence of global failure modes depends on the presence of wall-to-wall and floor-to wall connections and the presence of an adequately rigid and resistant floor [20].

Therefore, the first setup aims at estimating the local in-plane response of the floor while the second setup aims most at assessing global mode shapes by spreading the sensors inside the building. The paper has the following objectives: (i) the authors estimate the in-plane low-vibration structural response of a typical floor of masonry buildings made of steel

joists and brick tiles. An equivalent beam element is updated using the experimental modal parameters to estimate both the equivalent elastic modulus of the floor and its thickness; (ii) The in-plane low-vibration structural response of the entire building using the experimental modal parameters from the second setup and the mechanical parameters of the floor assessed in the first optimization step is deepened. The authors numerically evaluate the dynamic behaviour of the building by modelling with an equivalent frame model, the most approach most commonly applied by practitioners; (iii) Lastly, the influence of the floors' stiffness on the global dynamic behaviour of the selected building is better deepened through incremental dynamic analysis by varying a suite of mechanical values according to the variability observed among buildings in central–southern Italy. Fragility curves are derived, aiming at providing a reliable prediction of masonry buildings' seismic performance, which is also useful to foresee strengthening interventions required for floors.

## 2. Description of the Prototype Building

The authors' research aims to delve into the dynamic behaviour of ordinary residential facilities of the central–southern Italian built-up area. With this aim, a building was selected as an archetype of the ordinary masonry heritage, expressing its main features. The construction has a nearly rectangular shape, 10.06 m long and 12.23 m wide, and consists of two stories, 3.05 m high. Figure 1 reports the plans of the stories and Figure 2 depicts two views of the building. The bearing structure consists of irregular stone masonry units with good mortar, recently restored with grout injections, see Figure 3. The longitudinal direction consists of three alignments of resistant walls; the transverse one consists of four alignments of walls, of which the internal ones are built with clay bricks. The floors are made of steel joists and brick tiles. The openings show the presence of a lintel due to a good layout of masonry units and the presence of tension-resistant elements. The roof consists of one layer of timber planks and joists.

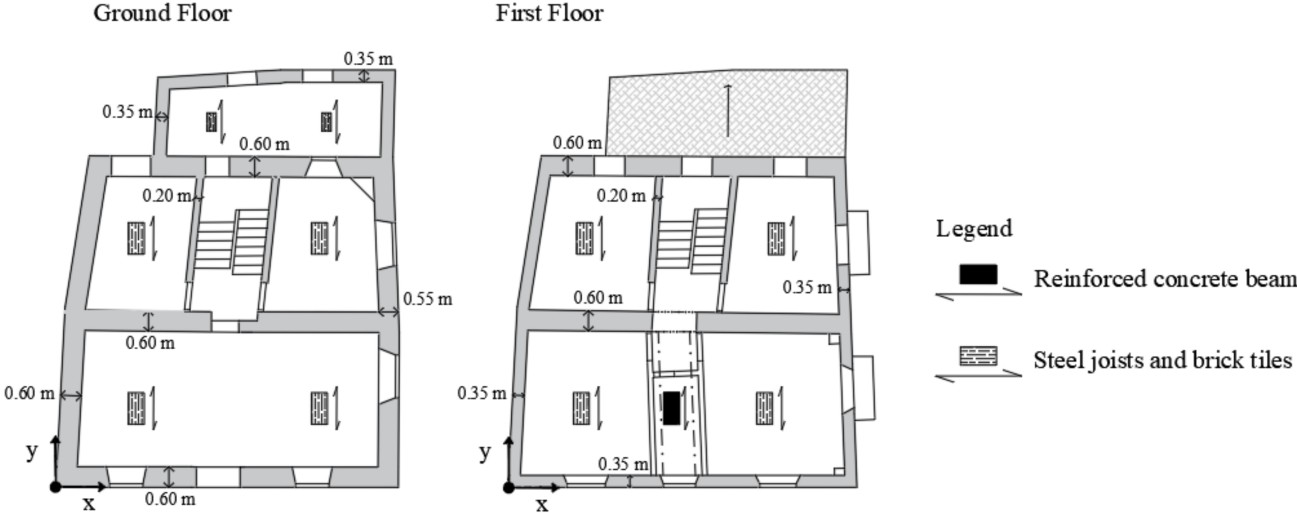

**Figure 1.** Plans of the building with the indication of the warping of horizontal structures.

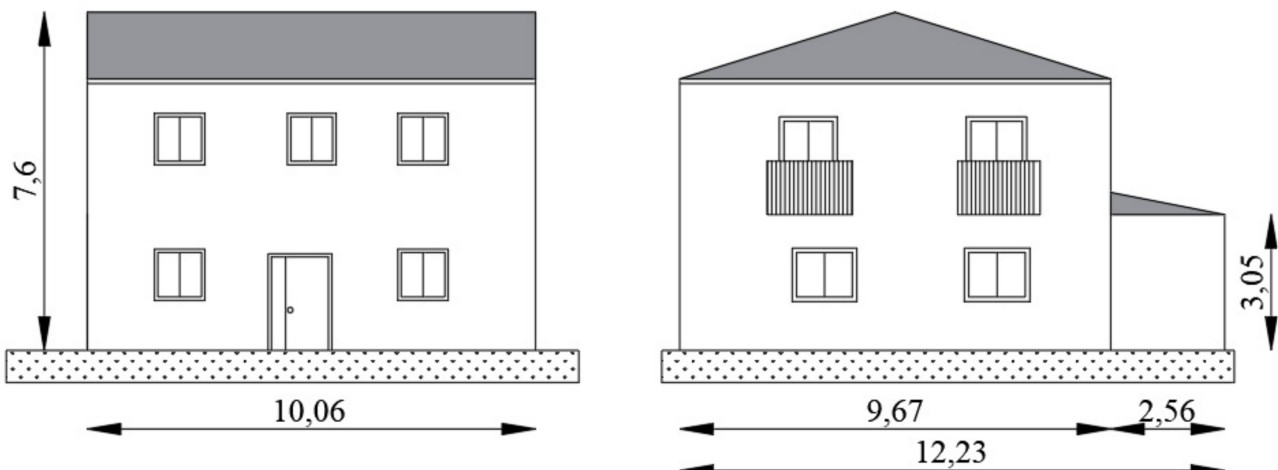

**Figure 2.** Prospects of the building with measures expressed in meters.

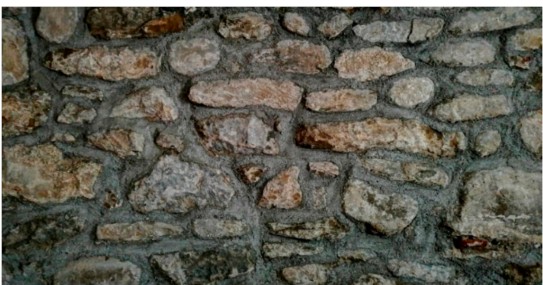

**Figure 3.** Texture of the masonry.

## 3. Numerical Modelling Description

Numerical analyses were carried out to assess the dynamic response of the building. Specifically, the authors developed finite element (FE) models implemented in the software package SAP2000. The first FE model aims to reproduce the local behaviour of the floors: a single equivalent beam is introduced with a section equal to the slab portion investigated by the first experimental setup, see Figure 4. In the middle of the equivalent beam, a portion of reinforced concrete models the presence of a flat beams, indicated in purple in Figure 4. The connection with the masonry walls was included in the model introducing linear springs characterized by the stiffness of the supporting walls.

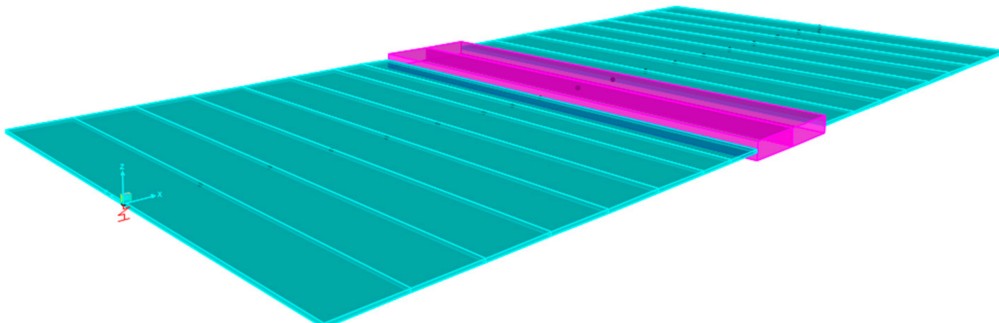

**Figure 4.** View of the numerical model of an equivalent beam representative of the floor system.

The second FE model reproduces the global masonry building, see Figure 5a,b. The masonry elements are represented using a continuum homogenized model with the finite element method, which ensures good results for the global analysis of masonry structures [21,22].

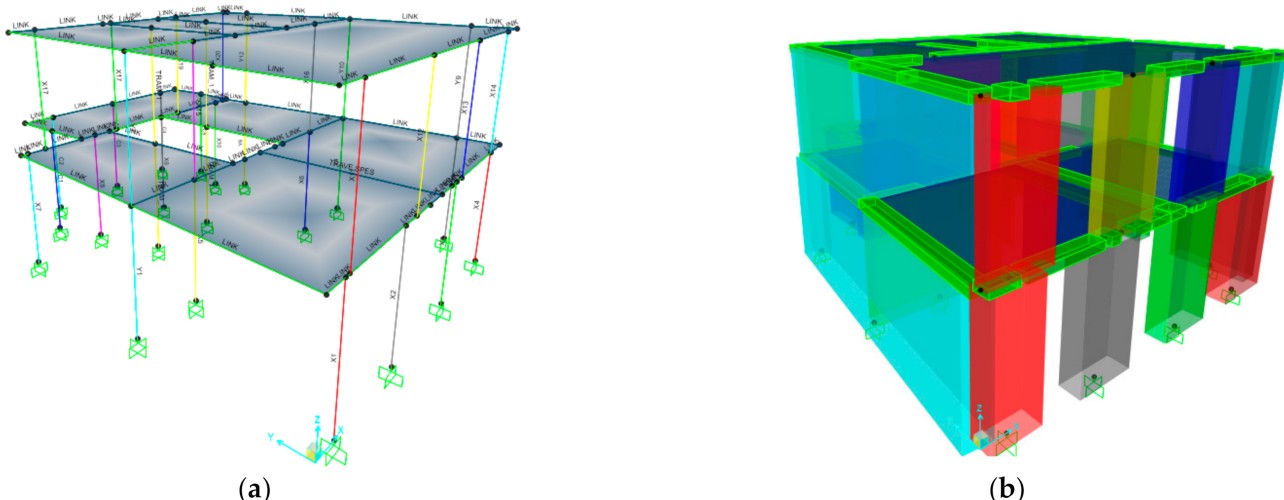

**Figure 5.** Numerical model of the global building: unifilar view (**a**); (**b**) extruded view.

The chosen approach consists of modelling masonry panels, namely, the piers (vertical elements) and masonry spandrels (horizontal elements) as non-linear beams, connected to each other by rigid links, see Table A1. Masonry portions confined between piers and spandrels are modelled as rigid nodes. The nonlinear mechanical behaviour derives from a so-called lumped plasticity approach [23]. The masonry elements are modelled as elastoplastic, with two rocking hinges located at the ends of each frame and a shear hinge located at the mid-height. The masonry piers may suffer in-plane failure for bending–rocking and shear sliding mechanisms. The constitutive law is represented in Figure 6, and the failure criteria follow the Eurocode 8 (EC8-1) [24] and the Italian Design Code [25,26]: the relationship 1 evaluates the ultimate moment of the rocking hinges and the relationship 2 evaluates the ultimate strength of the shear hinges.

$$M_u = 0.5\sigma_0 tD^2 \left(1 - \frac{\sigma_0}{0.85f_d}\right) \tag{1}$$

$$V_u = \frac{1.5f_{v0}tD}{\epsilon} \sqrt{1 + \frac{\sigma_0}{1.5f_{v0}}} \tag{2}$$

where $\sigma_0$ is the mean vertical stress for gravitational loading; $D$ and $t$ are the width and the thickness of the wall, respectively; $f_d$ is the design compression strength, $f_{v0}$ indicates the design shear strength with no axial force, and $\varepsilon$ is a coefficient related to the element geometrical ratio, assumed as H/D, where H is the height of the vertical masonry element.

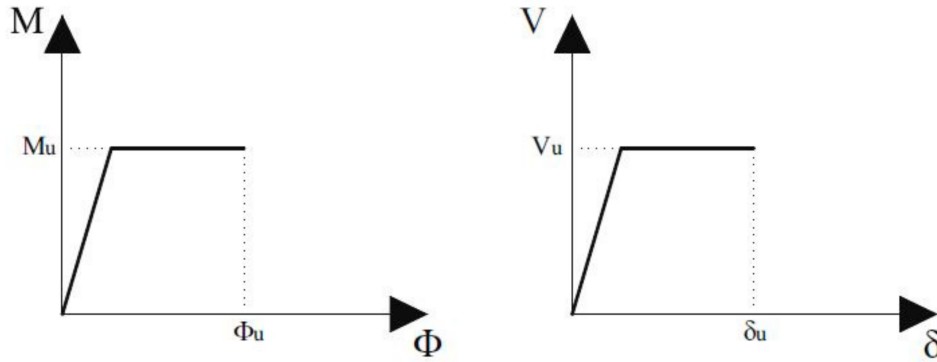

**Figure 6.** Constitutive laws of plastic hinges.

The ultimate shear displacement is equal to 0.4% of the deformable height of the masonry element, and the ultimate rotation for the bending moment is 0.6%.

The mechanical parameters of the masonries were assigned according to the Italian Standard Code, see Table 1. The masonry walls composed of stone units recently underwent a restoration with grout injections. As suggested by the Italian Standards, the mechanical characteristics are corrected by a correction factor equal to 1.7 to account for the effectiveness of the strengthening interventions on the mechanical characteristics of the masonry. This value of the elastic modulus $E_m$ provided the starting value in the model updating process. Moreover, the Confidence Factor, FC, equal to 1.35, corresponding to a limited knowledge level, LC1, was assigned to penalize the compressive and shear strength of the masonry, called $f_m$ and $f_{v0}$, respectively.

**Table 1.** Mechanical parameters of the masonry.

| Masonry Typology | Compressive Strength Average Value $f_m$ (MPa) | Shear Strength Average Value $f_{v0}$ (MPa) | Young Modulus Average Value E (MPa) | Shear Modulus Average Value G (MPa) | Specific Weight $\gamma$ (kN/m3) |
|---|---|---|---|---|---|
| Irregular layout with masonry units embedded with good mortar | 2.00 | 0.042 | 1230 | 460 | 13–16 |
| Reinforced with grout injections | 3.57 | 0.07 | 2754 | 918 | 13–16 |
| Clay bricks | 2.60 | 0.050 | 1500 | 500 | 18 |

The floor was modelled as an orthotropic membrane, with the elastic moduli, $E_{s1}$ and $E_{s2}$ of the slab, different for the main direction and the perpendicular direction, respectively [27,28]. The loads of the floors and roof were considered in the model as vertical concentrated forces. The structure was assumed to be fixed to the foundation.

Table 2 reports the main geometrical and mechanical characteristics of each of the masonry piers, as reported in the legend in Figure 7, that were implemented in the numerical model.

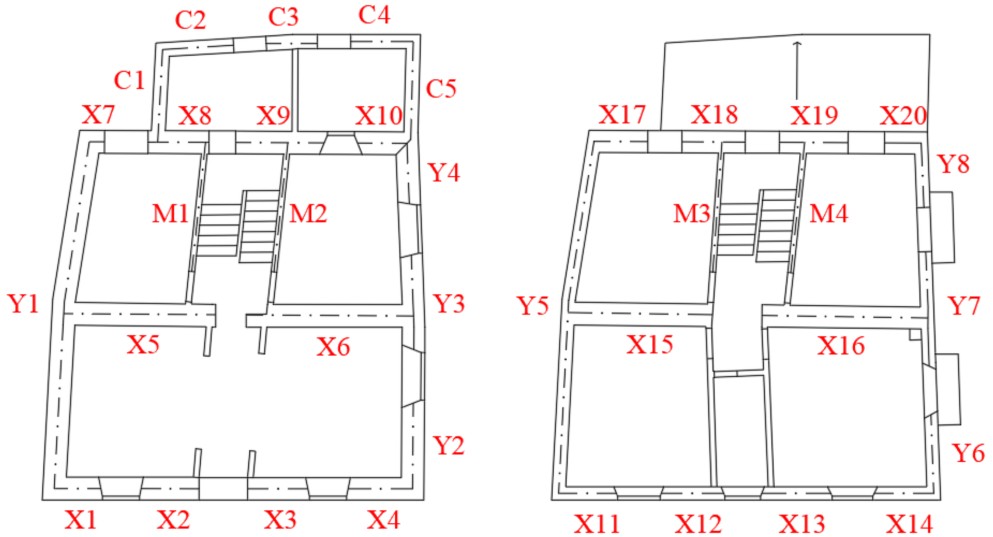

**Figure 7.** Legend of masonry piers.

**Table 2.** Identified modal parameters from the first setup.

| Mode | Frequencies (Hz) | Damping Ratio (%) |
| --- | --- | --- |
| 1st | 8.85 | 6.49 |
| 2nd | 9.43 | 2.17 |
| 3rd | 12.26 | 2.90 |

## 4. Ambiental Dynamic Identification: Description and Results

The authors carried out dynamic investigations using Operational Modal Analysis, OMA [29]. The analysis used ten force-balance mono-axial accelerometers, named SARA Instruments SA10, characterized by a dynamic range higher than 165 dB in the frequency interval 0.1–20 Hz and a 5 V/g sensitivity. Two measurement chains, depicted in Figure 8, were used for the acquisition system, each one driven by a master recorder unit and synchronized by a GPS sensor, see Figure 9b. The first measurement chain consisted of ten accelerometers located along a row, spaced every 1.05 m, see Figures 8a and 9c. In the second measurement chain, four couples of accelerometers were placed at the four corners of the building at the first floor, and the remaining couple was located at the middle of the floor, see Figures 8b and 9a.

The first setup is used to characterize the global dynamics of the structure, while the second is used to identify with high accuracy the mode shapes characterized by a prevalent deformation of the floor.

The aim of the first setup was the investigation of the modelled characteristics of the building, whereas the second setup deepened the understanding of the structural behaviour of the floor.

The data were sampled at a rate of 200 Hz and the cut-off frequency of the anti-aliasing filter was fixed to 20 Hz. The identification was carried out under environmental conditions.

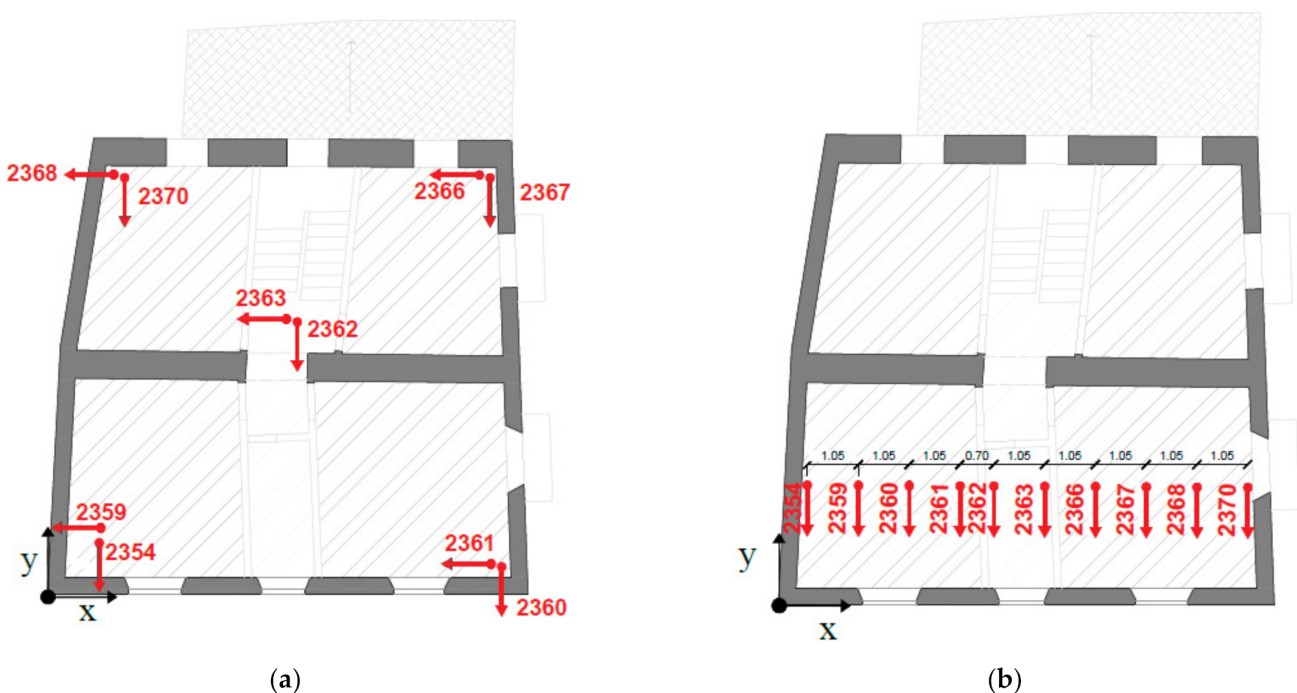

(a)                                                             (b)

**Figure 8.** Illustration of the two setups with the measuring direction (indicated by the red rows): (**a**) the first setup; (**b**) the second one.

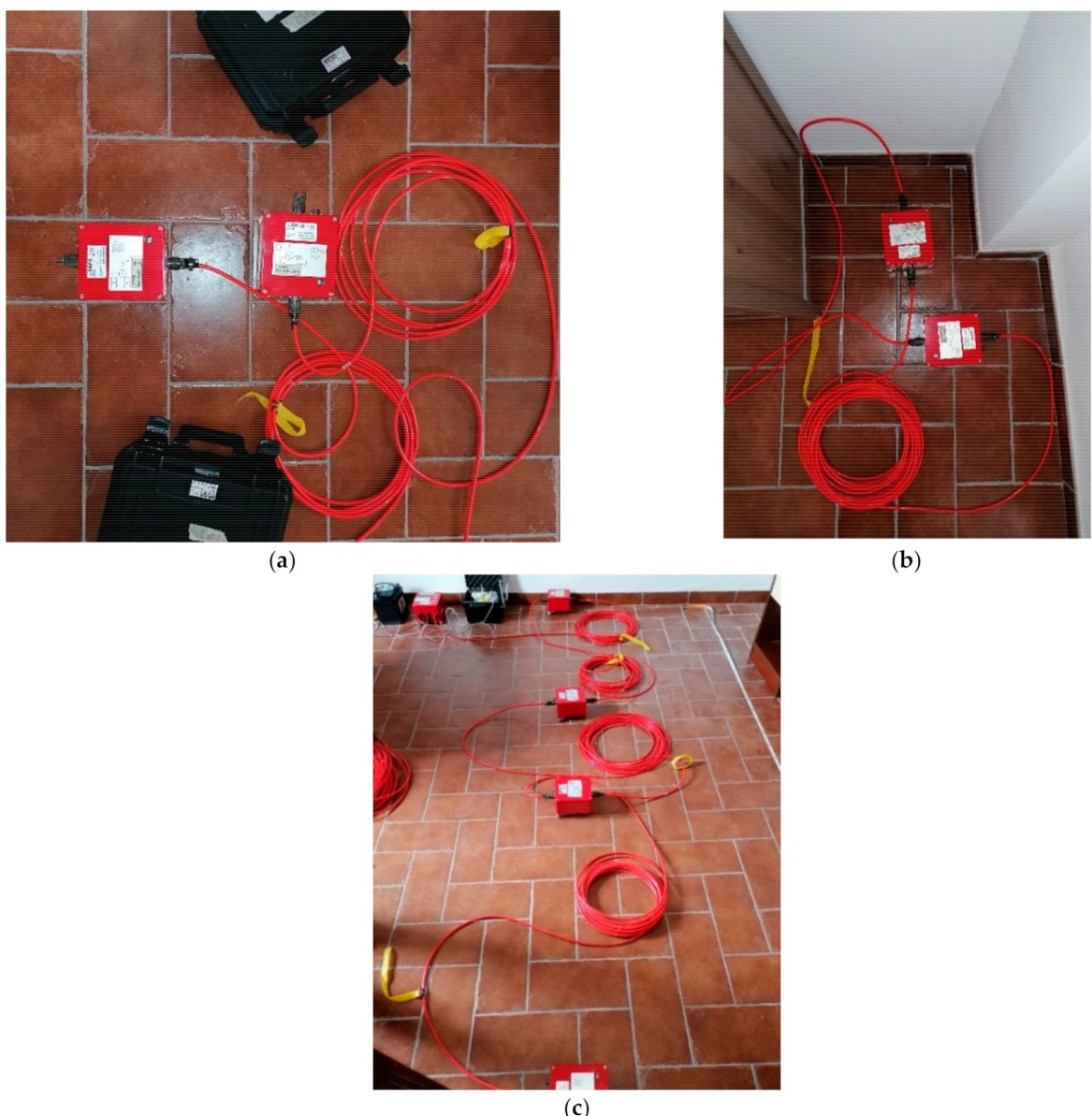

**Figure 9.** Views of the position of the accelerometers: (**a**) at a corner of the building; (**b**) at the middle of horizontal structure; (**c**) along a row.

The authors estimated the modal parameters from output-only measurements based on the covariance-driven stochastic subspace identification (SSI) method [30,31].

The SSI technique is a classical covariance-driven stochastic realization algorithm, namely the principal component algorithm, also known as covariance-driven SSI algorithm (SSIcov). Stabilization criteria were imposed to graphically verify the stability of the poles as the order of the system increased, see Equations (3)–(5):

$$\left| \frac{f_i - f_{i-1}}{f_{i-1}} \right| \leq \delta_f \tag{3}$$

$$\left| \frac{\xi_i - \xi_{i-1}}{\xi_{i-1}} \right| \leq \delta_\xi \tag{4}$$

$$1 - MAC(\Phi_i, \Phi_{i-1}) \leq MAC_{thr} \tag{5}$$

where $f_i$, $\xi_i$, and $\Phi_i$ are the natural frequency, damping, and mode shape registered for each pole of the i-th order, $I = n_{min} + 1, \ldots, n_{max}$, from the i-th iteration, respectively; the modal

assurance criterion (*MAC*) threshold value is $MAC_{thr} = 0.02$; $\delta_f = 0.01$, and $\delta_\xi = 0.01$ are the adopted tolerances for the natural frequencies and damping ratios.

Three modes arose clearly from the stabilization diagram, which is reported in Figure 10.

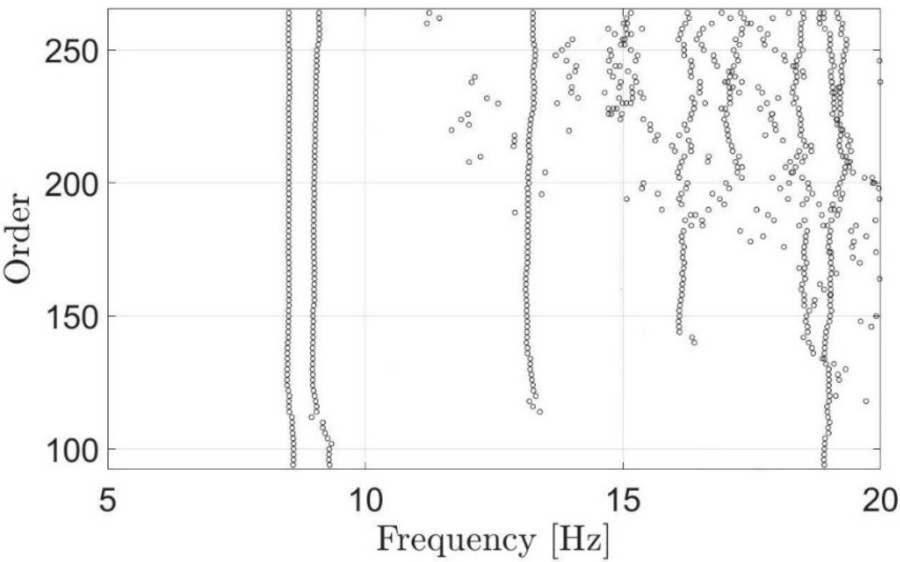

**Figure 10.** Stabilization diagram obtained from SS-I cov analysis on the first setup.

Table 2 summarizes the identified modal parameters.

The experimental mode shapes from the first setup are depicted in Figure 11.

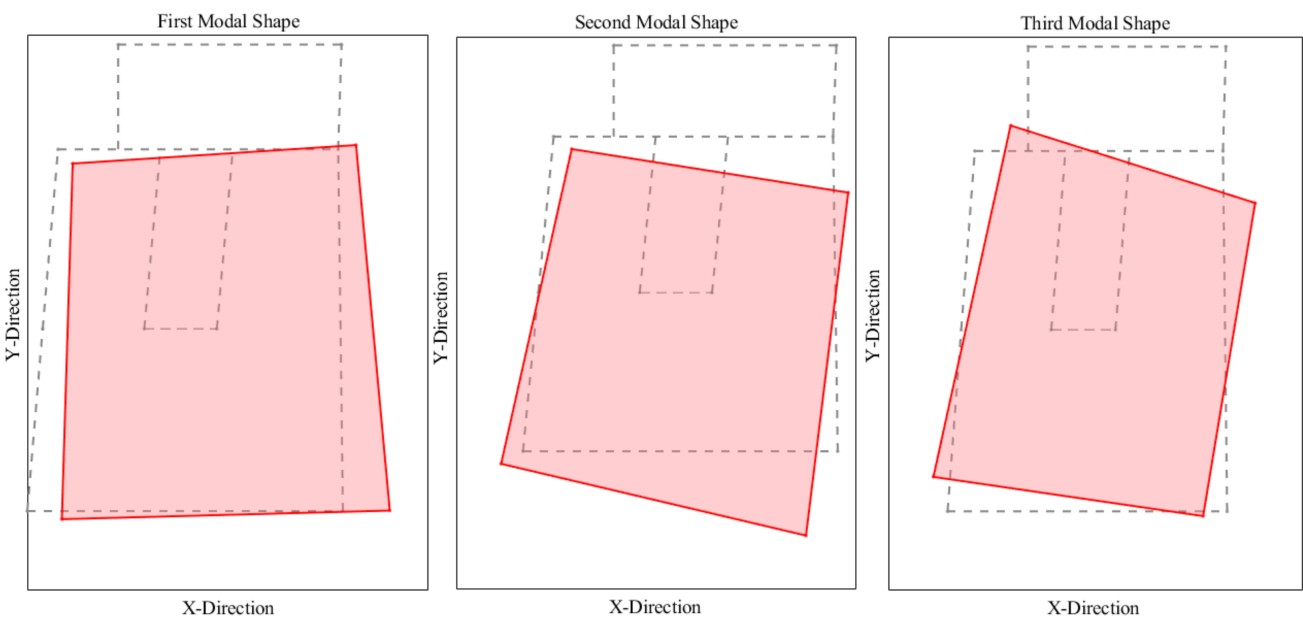

**Figure 11.** Experimental mode shapes.

## 5. Model Updating Process

The model updating process consists of the calibration of the finite element model of a structure to match numerical and experimental results [32] to obtain reliable predictions of structural behaviour. The model updating bases on the solution of a constrained optimization problem related to an objective function. Generally, the objective function involves

the difference between the experimental and numerical natural frequencies and the mode shapes of structures.

The authors selected the following objective function, see Equation (6), that measures the distance between the estimated modal parameters and the numerical ones [33].

$$C = \sum_{i=1}^{M} k_i \left( \frac{\omega_i{}^m - \omega_i{}^c}{\omega_i{}^m} \right)^2 + \eta \sum_{i=1}^{M} (1 - diag(MAC(\varnothing_i{}^m, \varnothing_i{}^c))) \tag{6}$$

where the apex $(*)^m$ indicates a measured variable and the apex $(*)^c$ a calculated variable; $\Phi_i$ is the mode shape vector; $M$ is the number of modes; $MAC$ is the modal assurance criterion [34]; and $k_i$ and $\eta$ are the weighting factors, set equal to 1.00.

In order to contain the effort of the model updating process, a parametric analysis was performed to address the selection of the parameter for the calibration of the FE models.

In the following subsection, the authors first carry out the sensitivity analyses to assess the influence of the mechanical parameters of both the horizontal and vertical structural elements on the global dynamics (Section 5.1). The analyses revealed that the modal parameters of the floor do not influence the global dynamics in the considered ranges of the parameters. Therefore, this evidence allowed a separate model updating of the local dynamics of the floor (Section 5.2) and the global dynamics of the entire structure (Section 5.3).

### 5.1. Sensitivity Assessment of the Dynamic Behaviour of the Structure in the Low-Amplitude Vibration in Operational Conditions

The authors investigated the dynamic behaviour of the selected building by varying mechanical properties of masonries and floors. The parametric analysis was based on four mechanical parameters: (i) the elastic modulus of the masonry composed by irregular masonry units, $E_m$; (ii) the elastic modulus of the masonry composed by clay bricks, $E_b$; (iii) the elastic modulus of the floors for the main direction, $E_{s1}$; and (iv) the perpendicular direction, $E_{s2}$. Acceptable ranges are investigated for each parameter, as recommended by the Normative Code and research studies of the literature [35], see Table 3.

**Table 3.** Ranges of values considered for the sensitivity analysis.

|          | Minimum Value [MPa] | Maximum Value [MPa] | Mean Value [MPa] |
|----------|---------------------|---------------------|------------------|
| $E_m$    | 2927.4              | 5854                | 4390.7           |
| $E_b$    | 1500                | 3000                | 2250             |
| $E_{s1}$ | 5000                | 25,000              | 17,500           |
| $E_{s2}$ | 5000                | 25,000              | 17,500           |

Numerical analyses of the buildings were carried out by varying the mechanical properties in the ranges indicated. The results are expressed in terms of frequencies and modal shapes to grasp the influence on the linear dynamic behaviour.

The parametric analysis highlights that the natural frequencies and the modal shapes of the building are mainly affected by the elastic moduli of masonries.

Therefore, the model updating process consisted in two phases: (i) the first model updating focused on the local behaviour of the building to grasp the horizontal structural influence, involving the elastic modulus $E_{s1}$ of the slab in the in-plane direction, the elastic modulus $E_{s2}$ in the out-plane direction and the section height of the slab; (ii) the second model updating involved the mechanical parameters of the masonry, the elastic modulus $E_m$ and the specific weight $\gamma_m$, focusing on the global behaviour of the building.

### 5.2. Model Updating of the local Behaviour of the Structure

The modal parameters estimated from the second setup are used for the model updating of the floor model.

The elastic modulus $E_{s1}$ and $E_{s2}$ of the slab, referring to the in-plane direction and the out-of-plane direction, respectively, were assigned according to the data from the diagnostic campaign on local built-up areas.

The range of both the elastic moduli was equal to 5000–25,000 MPa. The height of the section, H, ranges from 0.025 m to 0.05 m.

The optimization process was carried out considering the function in Equation (6) as the objective function to minimize, and returned a value of 8250 MPa for $E_{s1}$, 7500 MPa for $E_{s2}$, and the value of 0.025 m for the height of the slab section. The experimental frequency of the first mode was equal to 8.85 Hz; the numerical one, after updating the model, was equal to 8.83 Hz.

The MAC value between the experimental and numerical mode is s equal to 0.84, proving a consistent correspondence. The mechanical and geometric characteristics found were introduced in the global finite element model before carrying out the second phase of updating the model.

*5.3. Model Updating of the Global Behaviour of the Structure*

The modal parameters estimated from the first setup are used to update the global structural model.

An acceptable range for the values of the elastic moduli of the masonry was set and investigated.

The minimum values for the variation ranges of the mechanical parameters were set in accordance with the Italian guidelines. Based on the scatter of the mechanical parameters of the irregular stones' material, the maximum values are set slightly higher than the normative vales.

The minimum value of the Elastic Modulus of the masonry was set to 2000 MPa and the maximum values was set equal to 6000 MPa arbitrarily; the elastic modulus $E_b$ of the two bricks alignments ranges between 1200 to 2000 MPa. The specific weight of the irregular masonry units $\gamma_m$ varies into the range 12–24 kN/m$^3$.

As in the first parametric identification, the chosen objective function was that given in Equation (6).

The comparison between the numerical and experimental frequencies of the updated model is reported in Table 4.

**Table 4.** Ranges of values considered for the sensitivity analysis.

| Mode | Experimental Frequencies (Hz) | Numerical Frequencies (Hz) | Discrepancy (%) | Experimental Damping Ratio (%) |
|------|------|------|------|------|
| 1st | 8.85 | 8.53 | 3.61 | 6.49 |
| 2nd | 9.43 | 9.30 | 1.37 | 2.17 |
| 3rd | 12.26 | 13.25 | −8.07 | 2.90 |

The optimization process led to a value of 3000 MPa for $E_m$, 1980 MPa for $E_b$, and a specific weight of the masonry material $\gamma_m$ equal to 13 kN/m$^3$.

The estimated value of the specific weight is very low compared to the expected values for masonry, beyond 18 kN/m$^3$. This finding reveals that traditional masonry buildings exhibit a significant scatter of the mechanical properties of existing masonry. Specifically, traditional masonry can be characterized by high porosity, evidence of poor masonry quality despite the apparent good texture of the exposed faces.

Figure 12 reports the contour plot of the objective function for the parameters of masonry composed of irregular masonry units.

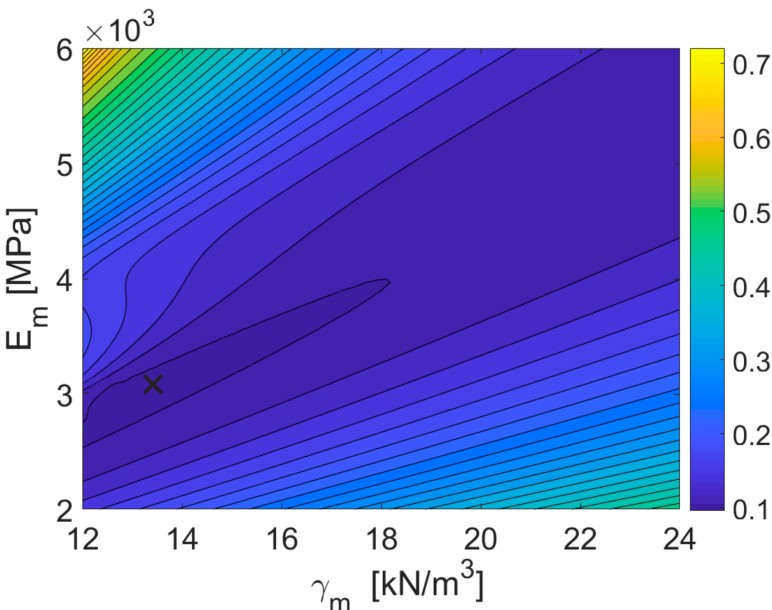

**Figure 12.** Contour plot of the objective function C in the range of variation of parameters $E_m$ and $\gamma_m$, at the identified value of $E_b$.

The mode shapes revealed by the numerical analysis after model updating process are depicted in Figure 13.

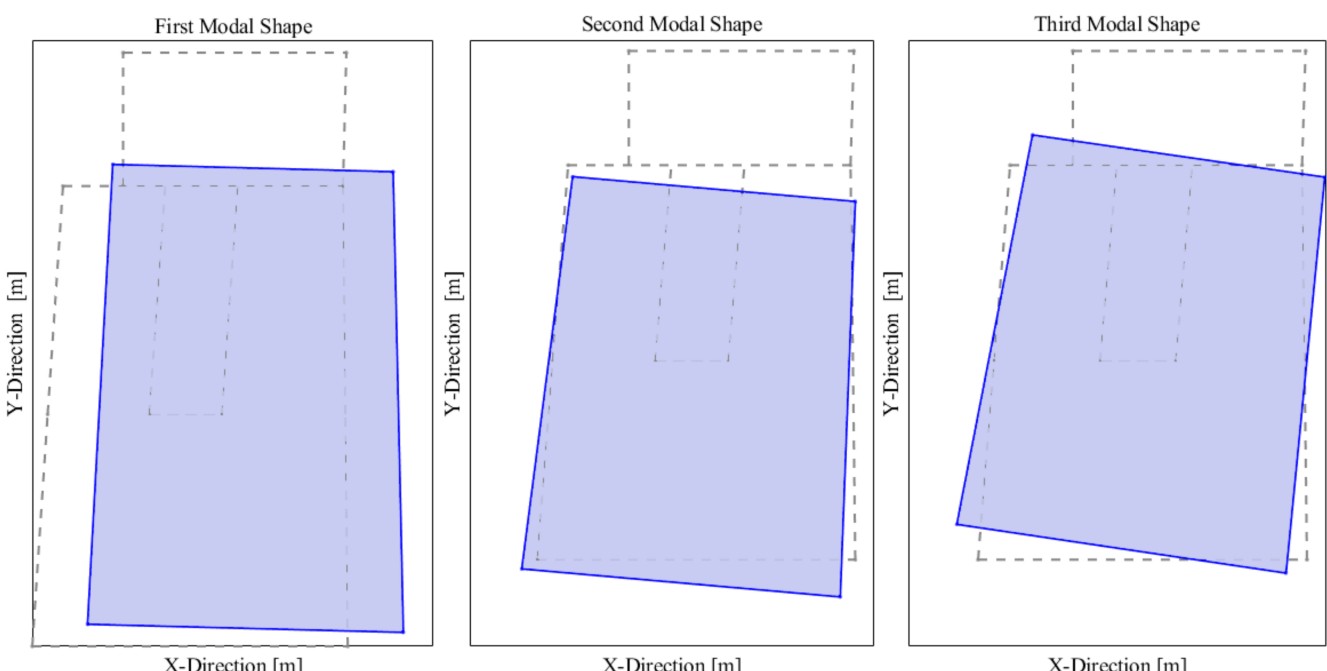

**Figure 13.** Numerical mode shapes after the model updating.

Figure 14 reports the MAC values evaluated for the numerical and experimental modes: the diagonal values demonstrate a satisfactory correspondence.

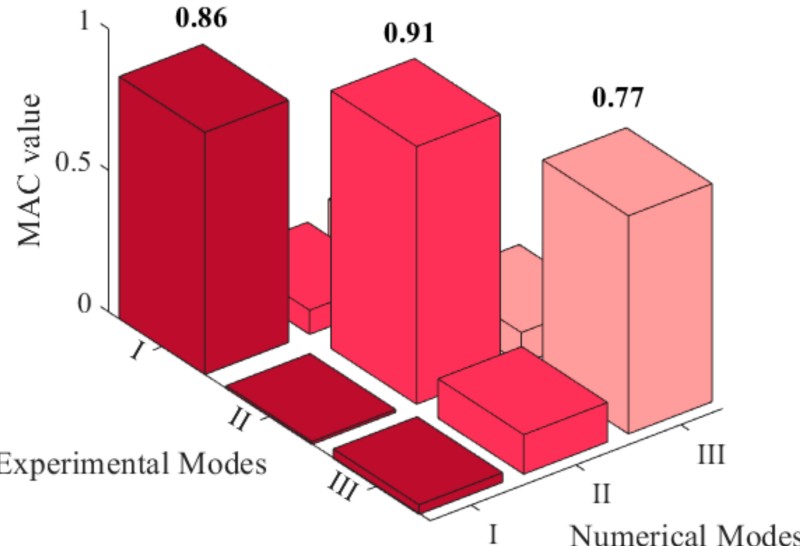

**Figure 14.** MAC representation between experimental and numerical modes.

In conclusion, the frequencies and MAC values in Figures 13 and 14 derive from the modal identification of the global measurement setup. The modal identification in Section 5.2 aimed at describing the local mode shape of the floor. The sensitivity analyses revealed that the floor's mechanical parameters do not influence the global structural dynamics. Therefore, these parameters were not used in the model updating of the global modes.

## 6. Numerical Investigation of the Structural Role of the Horizontal Structure in Non-Linear Dynamic Field

The updated FEM model is used to assess the effect of the floor stiffness on the seismic response.

The actual floor of the building is made of steel joists and brick tiles. The authors estimated the equivalent elastic modulus of other two floor typologies: a wooden floor consisting of timber joists supporting a timber deck and a traditional RC floor with hollowed tiles. Figure 15 depicts the three structural cross sections of floor typologies used in the proposed comparative assessment. Figure 16d is the photo of the floor of the selected building.

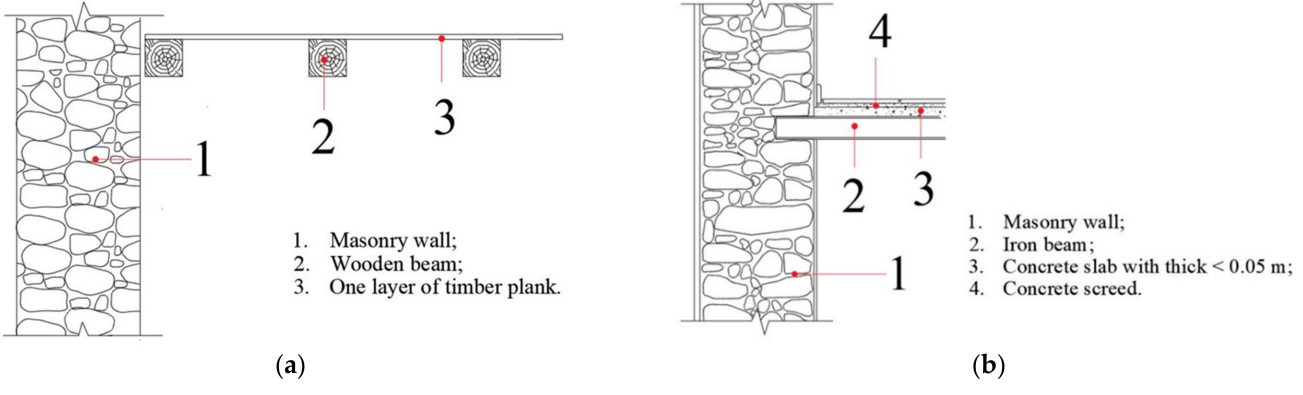

(a)  (b)

**Figure 15.** *Cont.*

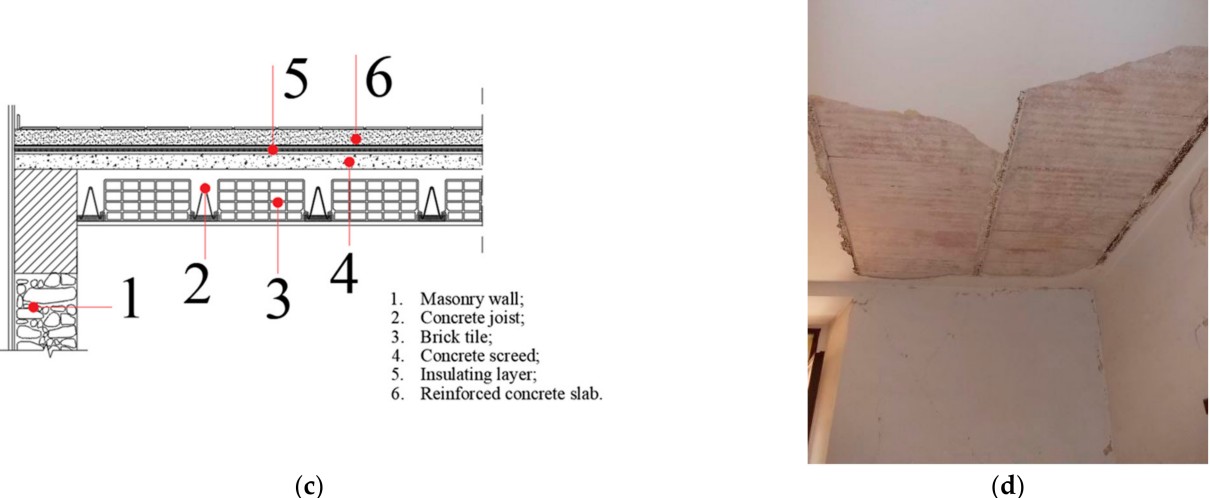

**Figure 15.** Floor typologies: (**a**) Structural cross sections of the timber floor; (**b**) Structural cross sections of the steel joists and brick tiles floor; (**c**) Structural cross sections of the RC floor; (**d**) Photo of the investigated floor.

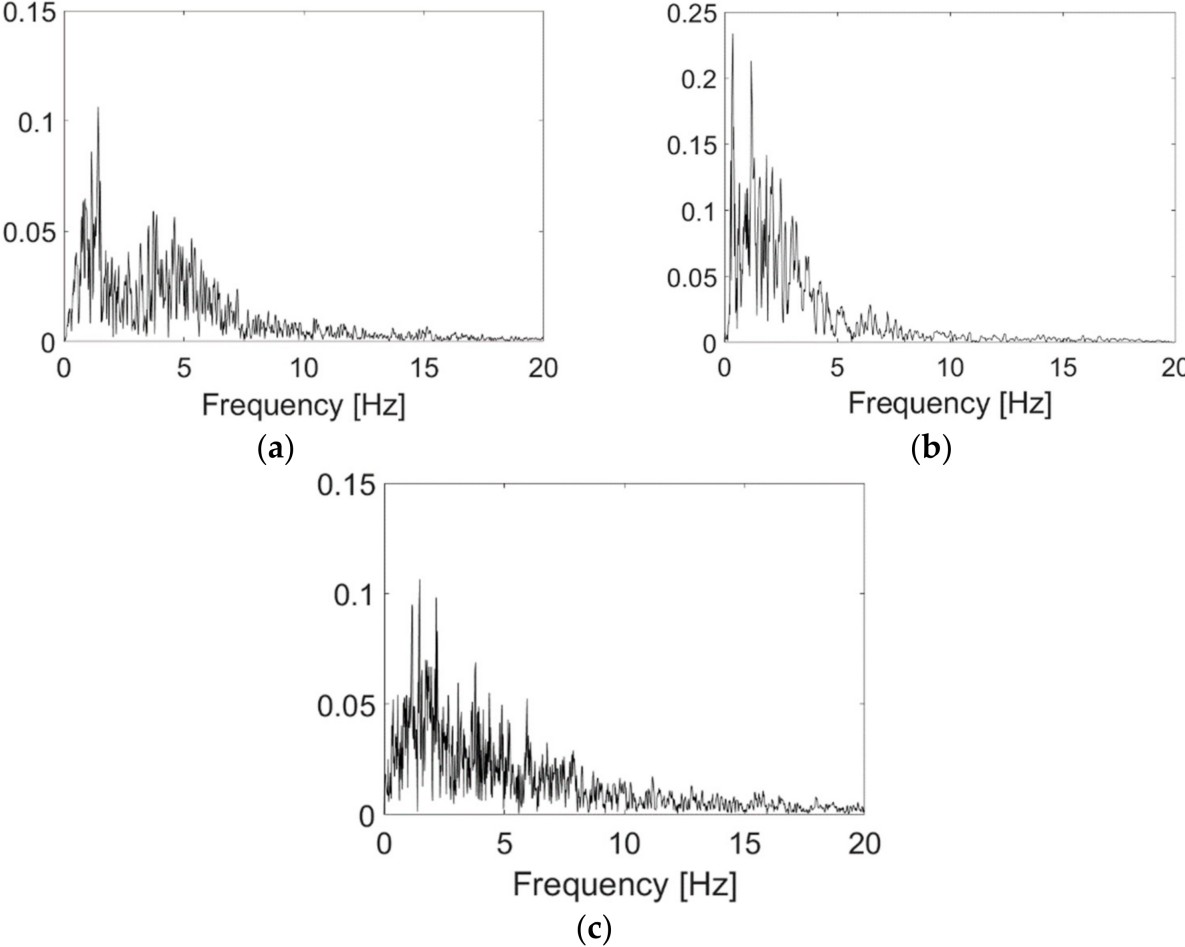

**Figure 16.** Frequency spectra: (**a**) L'Aquila earthquake; (**b**) El Centro earthquake; (**c**) Newhall earthquake.

Table 5 summarizes the main mechanical characteristics of the selected floor typologies.

**Table 5.** Mechanical characteristics of floor typologies.

| Floors Typology | Description | Young Modulus Value E (MPa) | Shear Modulus Value G (MPa) |
|---|---|---|---|
| HS1 | Investigated floor | 8250 | 3437.5 |
| HS2 | Wooden floor with joists and one layer of timber planks | 26 | 10 |
| HS3 | Reinforce concrete floors | 26,000 | 14,400 |

The estimation of the equivalent elastic modulus is based on a mechanical analogy. Specifically, the authors modelled three numerical models different for the typologies of floors: (i) HS1 is the label of the numerical model of the investigated building; (ii) HS2 is the label of the numerical model with timber wooden floors; (iii) HS3 is the label of the numerical model with RC floors. The modelling approach of masonry panels is descripted in Section 3. Then, we estimated the equivalent elastic modulus of an equivalent slab characterized by evaluated thicknesses, in order to obtain the same in-plane stiffness of the refined FE model of the floor.

The choice of this mechanical equivalence aimed at lightening the computational burden of the nonlinear static analyses, by using a single equivalent shell element.

The purpose of this study is estimating the effect of the floor in plane stiffness on the seismic response of an exemplary case study.

The authors selected seven seismic events, listed in Table 6. The storey drift was chosen as the target parameter to estimate the structural performance.

**Table 6.** List of the earthquake selected for the incremental dynamic analysis.

| Name | Areas Affected | Year | $M_w$ | PGA [g] |
|---|---|---|---|---|
| El Centro | United States, Mexico | 1940 | 6.9 | 0.35 |
| Erzikan | Erzincan Province, Turkey | 1939 | 7.8 | 0.50 |
| New hall | Japan | 1995 | 6.9 | 0.67 |
| L'Aquila | Italy | 2009 | 6.3 | 0.66 |
| Northridge | Southern California, United States | 1994 | 6.7 | 0.50 |
| Loma Prieta | San Francisco, United States | 1989 | 6.9 | 0.65 |
| Parkfield | California, United States | 2004 | 6.0 | 0.49 |

Figure 16 reports the frequency spectra of three of the seven accelerograms.

As a first step, the analyses show the storey drifts under different earthquake multiplication factors from 0.1 to 1 in the X and Y directions, as shown in Figure 17.

Then, the outcomes of the nonlinear dynamic analyses are used to estimate the failure probability by comparing the estimated drifts with the target drifts associated with the exceeding of a given limit state, according to Equation (7) [36].

$$(\text{Extremely Severe damage}) \quad S_{d4} = D_u \tag{7}$$

where $D_u$ is the ultimate displacements, defined according to the normative dispositions.

The probability of collapse, Pc, descends from the estimation of the standard normal cumulative distribution function Φ, as expressed in Equation (8), used to fit the fragility function with data collected from non-linear dynamic analysis, [37]:

$$P(C|IM = x) = \Phi\left(\frac{\ln\left(\frac{x}{\theta}\right)}{\beta}\right) \tag{8}$$

where $P(C|IM=x)$ is the probability of structure collapse due to a ground motion $IM = x$, Φ indicates the standard normal cumulative distribution function (CDF), $\theta$ is the median of the fragility function (the *IM* level with 50% probability of collapse), and $\beta$ is the

standard deviation of the ln *IM*. The parameters $\theta$ and $\beta$ were estimated, following an iterative procedure [38], by varying the parameters until a certain likelihood function was maximized, see Equation (9).

$$\{\hat{\theta}, \hat{\beta}\} = \sum_{j=1}^{m} \left[ \ln \phi \left( \frac{\ln(IM_i/\theta)}{\beta} \right) \right] + \lceil n - m \rceil \ln \left[ 1 - \phi \left( \frac{\ln(IM_{max}/\theta)}{\beta} \right) \right] \tag{9}$$

where ^ denotes an estimated parameter, $\phi$ () is the standard normal distribution PDF, n the number of ground motion used in the analysis, m indicated the number of m ground motions that caused collapse at *IM* level lower than *IMmax*.

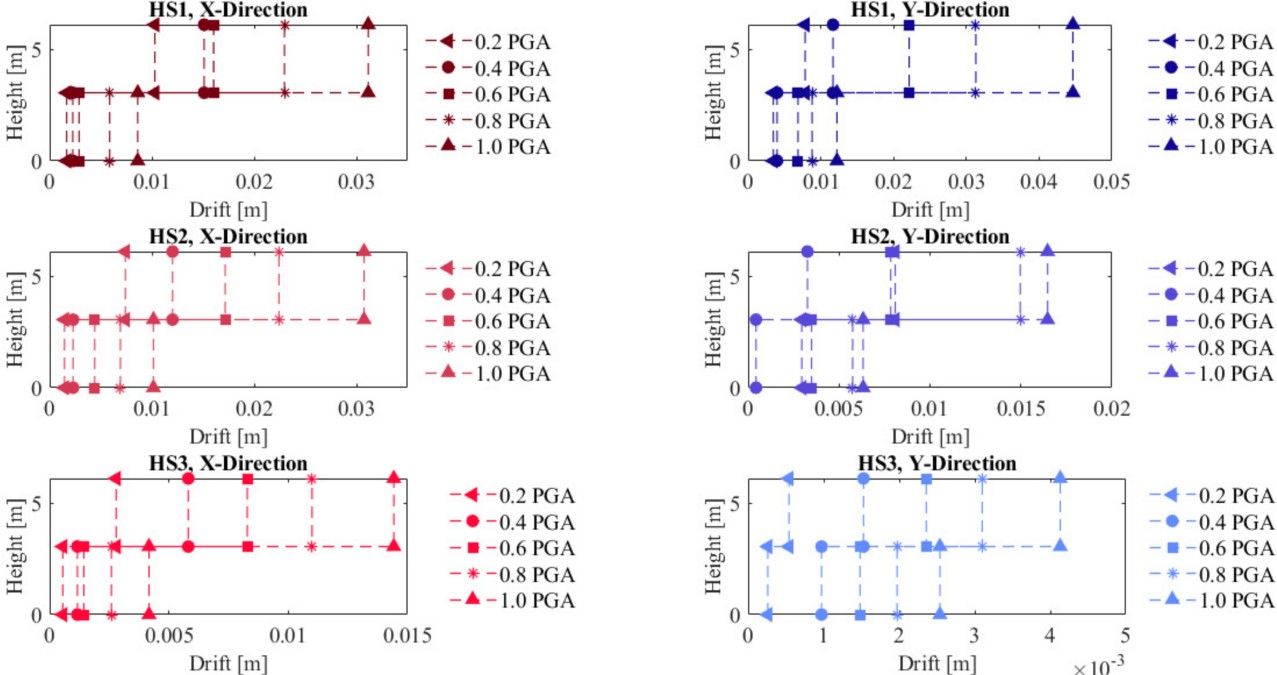

**Figure 17.** Storey drifts under different earthquake multiplication factors for the two main directions, at varying floor typology.

Figure 18 shows the mean drift values estimated from the response to the seven earthquakes under a specific peak ground acceleration, PGA, value. The X-direction is stiffer than the Y one due to the geometric configuration of the building, characterized by stiffer walls in the Y direction. The effect of the floor stiffness is the redistribution of the internal stresses between orthogonal walls. There is no marked difference between the drift and displacement response in the X direction using the HS1 and HS2 floors.

Conversely, the use of the HS2 floor produces a significant reduction in the storey drift in the Y direction, compared to the timber floor, higher than 200%. This increment of the storey drift value is evident at higher PGAs due to the activation of the plastic hinges. Therefore, the increment of the equivalent elastic modulus between HS1 and HS2 produces a more than 200% decrement of the storey drift. The stiffening of the floor by total percentage does not determine the modification of the response in the X direction, since the floor can be considered stiff due to the reduction in free-length in that direction. The comparison between HS1 and HS2 starts manifesting if the floor free length is higher than 4 m. If the floor free length is lower than 4 m, the effect of the floor typology does not arise, and the displacement associated with a timber floor does not differ from those related to the use of a traditional floor with steel joists and a RC slab with a thickness lower than 0.05 m. If the floor's free length exceeds 4 m, different floor typologies, even if both are considered not adequately stiff, produce a significant difference in the storey drift.

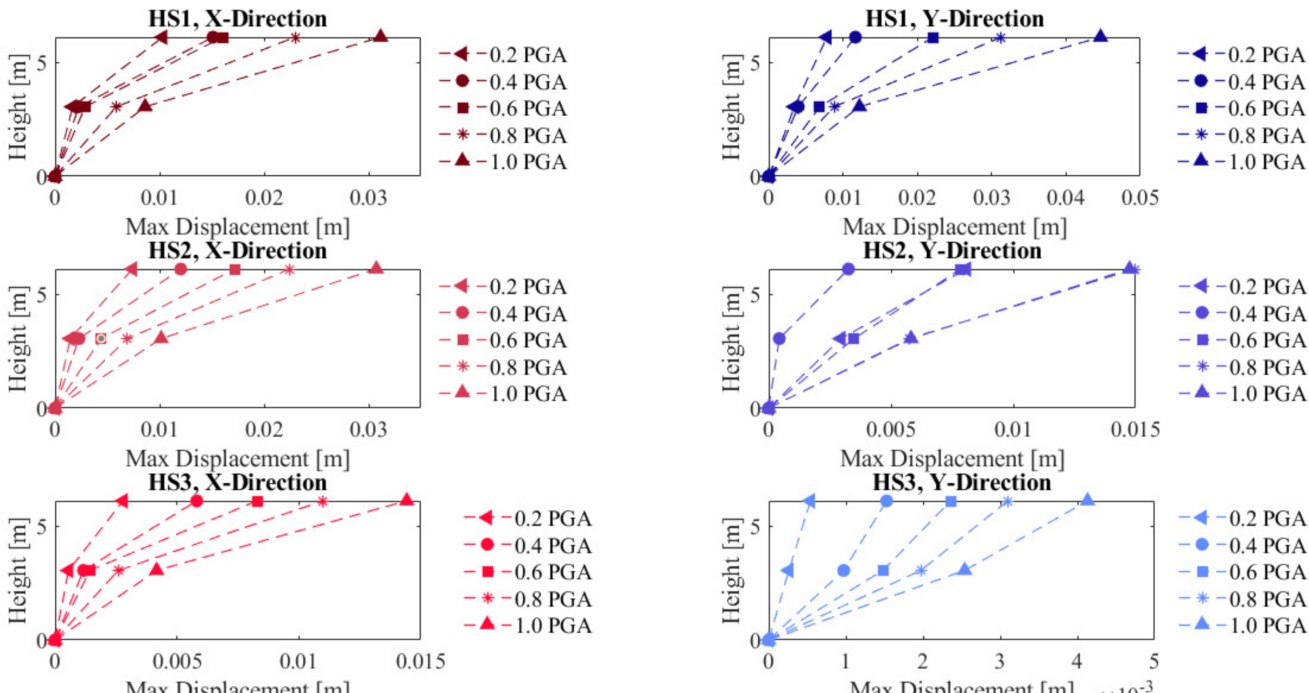

**Figure 18.** Mean drifts values under different earthquake multiplication factors for the two main directions, at varying the floor typology.

Conversely, using an RC floor made by a slab with more than 0.05 m thickness causes evident effects in both the X and Y direction. Specifically, the use of the HS3 floor determines a reduction in the storey drift equal to approximately 100% in both directions.

Despite the evident advantages of using rigid diaphragms, recent research shows that substituting traditional wooden floors with rigid diaphragms, i.e., RC floors, can cause unwanted consequences such as cracks on the edges of the two materials or, in the worst scenario, disintegration and collapse of the masonry walls [39].

The use of floors with an equivalent elastic modulus higher than this contributes to reducing the interstorey drifts in both directions, even if the floor free length is lower than 4 m in the X-direction. Next to reducing the storey drift, the use of floors with adequate in-plane stiffness, such as HS3, contributes to regularizing the displacement response along with the height, as evidenced by Figure 19.

The reduction in the storey drift related to using a floor with higher in-plane stiffness determines the reduction in the failure probability.

Figure 19 shows the fragility functions referring to the ultimate limit state of the considered building in the two directions.

Although the HS2 floor is considered not to be adequately stiff, it significantly reduces the collapse probability compared to HS1. Besides, as expected, the HS3 floor exhibits the most satisfactory seismic performance even at higher PGA values in both the X and Y directions.

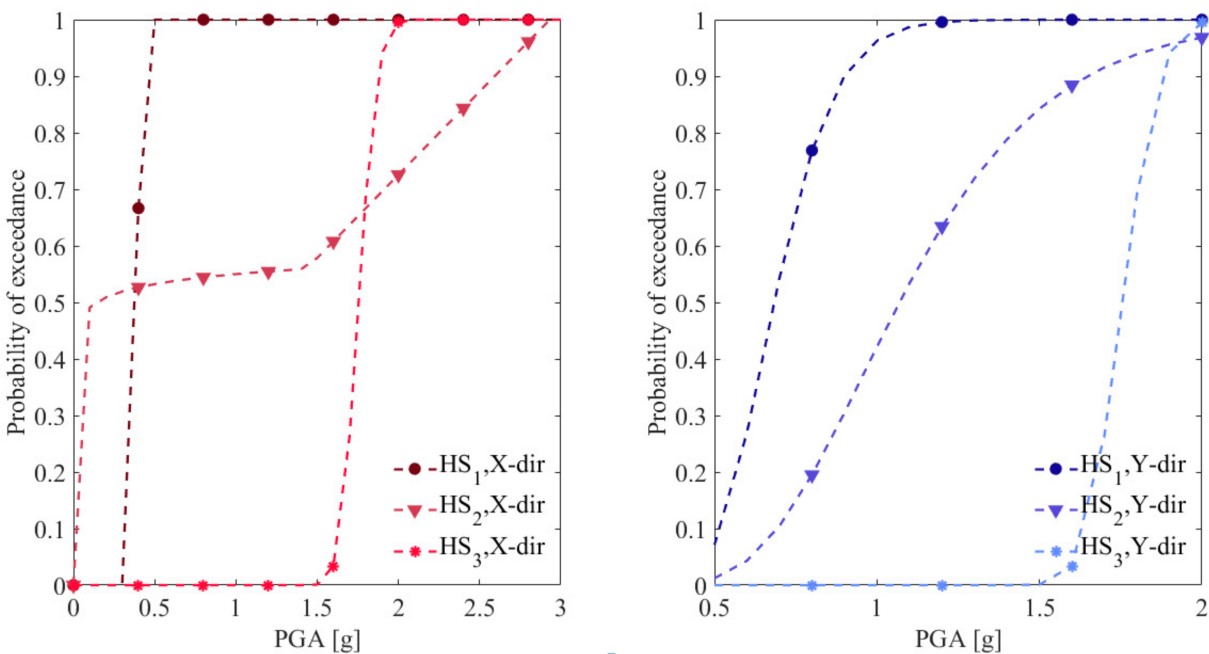

**Figure 19.** Fragility curves of buildings for the two main directions of the building, at varying the floor typology.

## 7. Conclusions

The seismic assessment of traditional masonry buildings is based on a conventional approach due to the lack of a solid predictive model of its structural response. Therefore, understanding their dynamic behaviour is crucial in research for a possible transition between conventional design approaches to ones with a more appropriate mechanical base. The current research aimed to delve into the dynamic behaviour of ordinary residential buildings to strengthen the knowledge about the seismic performance of existing architectural heritage through experimental and numerical analyses. The authors presented a combined application of Operational Modal Analysis (OMA) and model updating to estimate the modal parameters of a traditional masonry residential building.

With this aim, an in-depth experimental campaign was carried out through two ambient vibration tests of a selected building. First, a sensitivity analysis addressed the choice of the parameters to be involved in the model updating, reducing the numerical burden. The parametric analysis results evidenced the more marked influence of the mechanical characteristics of the masonry than of the floors on the linear dynamic response. Therefore, the authors separately updated the FE model of the floor and the entire structure using the experimental modal parameters estimated from the two setups. The diagonal MAC values of the updated model of the entire structure were equal to 0.86, 0.91, and 0.77, respectively, demonstrating the reliability of an equivalent frame model to grasp the global structural dynamics. In a second step, the authors assessed the influence of the floor on the non-linear dynamic behaviour of masonry buildings by varying a suite of mechanical values according to the variability observed among buildings in central–southern Italy: a wooden floor consisting of timber joists supporting a timber deck, HS1; a floor made of steel joists and brick tiles HS2, and a traditional RC floor with hollowed tiles, HS3. The displacements' trend, along with the height of the building, derived via non-linear incremental dynamic analysis, evidenced the notable role of the floor's stiffness in the non-linear dynamic behaviour of the building. There was no marked difference between the drift and displacement response in the X direction using the HS1 and HS2 floors, considering that the X-direction is stiffer than the Y one.

Conversely, the use of the HS2 floor produced a significant reduction in the storey drift in the Y direction, compared to the HS1 floor, higher than 200%. The stiffening of the floor did not modify the response in the X direction, since the floor could be considered stiff due

to its reduced free length. The comparison between HS1 and HS2 started manifesting if the floor free length was higher than 4 m. Instead, using an RC floor made by a slab with more than 0.05 m thickness caused evident effects in both the X and Y directions: a reduction in the storey drift equal to approximately 100% was revealed in both directions, even in the case of free length lower than 4 m in the X-direction, regularizing the displacement response along with the height, and thus reducing the failure probability. Lastly, the authors estimated fragility functions to evidence the influence of stiffness floors in terms of failure probability under a highly severe damage state. Although the HS2 floor was considered not to be adequately stiff, it significantly reduced the collapse probability compared to HS1. Besides, as expected, the HS3 floor exhibited the most satisfactory seismic performance, even at higher PGA values. The derived fragility curves reliably predicted masonry building's failure probability, also revealing helpful information allowing us to foresee the influence of strengthening interventions of floors on ordinary masonry buildings.

**Author Contributions:** Conceptualization, I.C., R.C. and A.A.; methodology, I.C., R.C and A.A.; software, I.C., R.C. and A.A.; validation, I.C., R.C. and A.A.; formal analysis, I.C., R.C. and A.A.; investigation, I.C., R.C. and A.A.; resources, I.C., R.C. and A.A.; data curation, I.C., R.C. and A.A.; writing—original draft preparation, I.C. and A.A.; writing—review and editing, I.C., and A.A.; visualization, I.C., R.C. and A.A..; supervision, M.F. and F.D.F.; project administration, I.C., R.C. and A.A..; funding acquisition, I.C., R.C. and A.A. All authors have read and agreed to the published version of the manuscript.

**Funding:** This research received no external funding.

**Institutional Review Board Statement:** Not applicable.

**Informed Consent Statement:** Not applicable.

**Data Availability Statement:** Data are available upon reasonable request.

**Acknowledgments:** The authors acknowledge the contribution of Tamara De Andreis for providing valuable information on the case study building analysed.

**Conflicts of Interest:** The authors declare no conflict of interest.

## Appendix A

**Table A1.** Geometrical and mechanical parameters of masonry piers.

| Masonry Piers | Width D (m) | Thickness t (m) | Height (m) | Tensional State $\sigma_0$ (MPa) | Ultimate Moment at the Top of Pier $M_u$ [kN·m] | Ultimate Moment at the Base of Pier $M_u$ [kN·m] | Ultimate Shear Strength $V_u$ (MPa) |
|---|---|---|---|---|---|---|---|
| X1 | 1.22 | 0.6 | 1.91 | 102.37 | 35.63 | 48.52 | 50.80 |
| X2 | 1.52 | 0.6 | 1.45 | 121.40 | 67.14 | 85.42 | 117.82 |
| X3 | 1.60 | 0.6 | 1.00 | 118.70 | 75.51 | 90.29 | 180.58 |
| X4 | 1.70 | 0.6 | 2.17 | 125.24 | 78.81 | 116.82 | 107.67 |
| X5 | 3.80 | 0.6 | 2.70 | 507.13 | 1259.86 | 1411.64 | 813.53 |
| X6 | 4.15 | 0.6 | 2.70 | 559.21 | 1575.55 | 1706.37 | 1010.72 |
| X7 | 0.60 | 0.6 | 2.70 | 97.95 | 8.69 | 10.88 | 8.06 |
| X8 | 1.60 | 0.6 | 2.70 | 136.38 | 74.05 | 112.66 | 83.45 |
| X9 | 2.33 | 0.6 | 2.70 | 170.25 | 183.19 | 249.73 | 205.10 |
| X10 | 1.72 | 0.6 | 2.70 | 118.60 | 70.24 | 119.83 | 88.76 |
| X11 | 1.52 | 0.35 | 1.90 | 44.29 | 12.60 | 21.92 | 23.07 |
| X12 | 1.65 | 0.35 | 1.00 | 51.72 | 20.56 | 26.76 | 53.51 |
| X13 | 1.82 | 0.35 | 1.00 | 51.97 | 24.76 | 33.07 | 66.14 |
| X14 | 1.55 | 0.35 | 2.10 | 46.60 | 13.40 | 24.28 | 23.13 |

**Table A1.** *Cont.*

| Masonry Piers | Width D (m) | Thickness t (m) | Height (m) | Tensional State $\sigma_0$ (MPa) | Ultimate Moment at the Top of Pier $M_u$ [kN·m] | Ultimate Moment at the Base of Pier $M_u$ [kN·m] | Ultimate Shear Strength $V_u$ (MPa) |
|---|---|---|---|---|---|---|---|
| X15 | 3.58 | 0.60 | 2.70 | 207.15 | 470.00 | 844.16 | 515.62 |
| X16 | 4.38 | 0.60 | 2.70 | 239.63 | 778.37 | 1414.58 | 810.97 |
| X17 | 1.42 | 0.60 | 1.90 | 49.76 | 17.72 | 39.85 | 41.93 |
| X18 | 1.67 | 0.60 | 1.00 | 49.46 | 30.26 | 49.22 | 98.43 |
| X19 | 1.80 | 0.60 | 1.00 | 57.86 | 41.94 | 65.35 | 130.7 |
| X20 | 0.95 | 0.60 | 2.10 | 37.97 | 6.21 | 13.67 | 13.02 |
| Y1 | 9.15 | 0.50 | 3.00 | 450.09 | 4648.35 | 6889.02 | 3368.74 |
| Y2 | 2.22 | 0.50 | 2.50 | 109.16 | 87.44 | 157.19 | 125.75 |
| Y3 | 2.45 | 0.50 | 1.00 | 69.21 | 78.06 | 118.27 | 236.55 |
| Y4 | 1.6 | 0.50 | 2.10 | 85.99 | 39.24 | 76.19 | 56.44 |
| Y5 | 9.29 | 0.35 | 2.70 | 175.14 | 972.47 | 3354.18 | 1918.73 |
| Y6 | 2.07 | 0.35 | 2.70 | 50.73 | 19.26 | 53.22 | 38.42 |
| Y7 | 3.02 | 0.35 | 2.10 | 27.12 | 0.00 | 83.02 | 79.07 |
| Y8 | 1.7 | 0.35 | 2.70 | 45.84 | 12.99 | 31.47 | 23.31 |
| M1 | 3.4 | 0.15 | 2.70 | 69.89 | 39.24 | 76.19 | 56.44 |
| M2 | 3.4 | 0.15 | 2.70 | 69.89 | 39.24 | 76.19 | 56.44 |
| M3 | 3.4 | 0.15 | 2.70 | 23.30 | 0.00 | 39.24 | 29.07 |
| M4 | 3.4 | 0.15 | 2.70 | 23.30 | 0.00 | 39.24 | 29.07 |
| C1 | 2.64 | 0.30 | 2.70 | 26.13 | 0.00 | 52.47 | 38.87 |
| C2 | 1.86 | 0.30 | 2.70 | 41.19 | 11.62 | 29.53 | 21.88 |
| C3 | 1.35 | 0.30 | 2.70 | 34.79 | 5.76 | 12.68 | 9.40 |
| C4 | 1.58 | 0.30 | 2.70 | 35.94 | 7.49 | 18.56 | 13.75 |
| C5 | 2.36 | 0.30 | 2.70 | 23.36 | 0.00 | 37.65 | 27.89 |

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
