# Peer review of "Operational Modal Analysis and Non-Linear Dynamic Simulations of a Prototype Low-Rise Masonry Building"

_buildings, doi:10.3390/buildings11100471_

Round 1
Reviewer 1 Report
Dear authors,
Thank you very much for the interesting research. The topic of the manuscript is in accordance with the Journal requirements. I pretty much enjoyed reading it. The manuscript is very well structured and critical parts are explained. I appreciate the effort for writing it in a proper manner.
The dynamic behaviour of masonry buildings in the Mediterranean region is the topic researched among numerous researchers, and the OMA method is not very well accepted. Probably the reason is the price of fieldwork. I appreciate the work.
Here are my recommendations and remarks.
- Formating
The manuscript is not in the template. Introduction should be 1 and Description of the prototype building number 2. Figure 3 in o the wrong place and covers the title of Chapter 3
- Title
Maybe it can be more specific – maybe you can add …numerical simulations o a prototype building (or case study)
- Abstract
ok
- Introduction
- Reference 1-4 – There is a lot more investigation on a full scale specimens, in example scholars from Slovenia, Croatia and Macedonia have many papers on the same topic. A lot of research was carried out in the IZIIS institute
- Destructive tests give insight into the mechanical properties of a limited area where the tests are executed. Conversely, dynamic tests may deliver comprehensive knowledge into the homogenized mechanical parameters by updating a finite element model using experimental modal parameters. I partially agree. I think NDT or semi-NDT is a must. Mechanical properties are of a limited area but still better than pure guessing of the material properties. Also, the procedure by Borri et al. gives a visual classification of the masonry properties quite well. Maybe it can be beneficial for the paper to compare the visual classification and recommendations from Italian norms. The second part of the sentence: may deliver comprehensive knowledge of homogenized mechanical… - dynamic test can give an insight into a global behaviour of the structure, not in the mechanical properties. Of course, it gives the indication, but the differences can be more than 100%. Maybe these sentences should be modified.
- OMA can return info about the elastic modulus – please explain how. I don’t understand how?
- Reference 11 – why this one?
- Numerical analyses
- Why wasn’t the building modeled with the true geometry? It’s simplified but from my point of view at point Y1 (figure 7) the behaviour will not be as in the model because of the geometry change.
- Table 1 – again, it can be compared to a visual classification (MQI method by Borri et al.)
- Table 2 -it should be in the appendix. σ0 is from SAP? Do you think this is a good or ie. Flat jack method should be applied to get the σ0? From our research it’s impossible to get correct σ0 from the model.
- Figure 8 – Why was the setup like this? Although I understand the concept, I will always recommend to add one more measuring point like on the drawing. The change in geometry influence the behaviour. This would be for sure seen in the Figure 11 if it was applied as I sketched.
- Model updating
- Table 4 – ranges of considered values seems quite high, especially for the examined building. Correlate it with some experimental research where NDT was used.
- Numerical investigation of the structural role of the horizontal structure in non-linear dynamic field
- The actual floor of the building is made of steel joists and brick tiles. Can you show the photo? Is it some prefab floor? It looks like standard prefabricated floors used in southern Europe in the last century.
- Table 6: HS3, HS1, HS5 – I don’t understand why numbers 3,1 and 5 were used? Is it a typo?
- Table 7 – you don’t need to change it, but it could be beneficial to have earthquakes from ie. Albania or Greece where this type of buildings is typical.
- Page 17: Specifically, the use of the HS3 floor determines a reduction of the story drift equal to approximately 100% in both directions. Maybe it does, but recent research suggests that replacing traditional wooden floors with rigid diaphragms, i.e., RC floors, can induce some unwanted consequences such as cracks on the edges of the two materials or, in the worst scenario, disintegration and collapse of the masonry walls (ie. https://doi.org/10.3390/su13116353 and a lot of Italian articles)
- Conclusion
- At this aim, an in-depth experimental campaign is carried out through two ambient vibration tests of a selected building selected.
- The comparison between HS1 and HS2 start manifesting if the floor free length is higher than 4 m. Is it common to have floor free length higher than 4 m in buildings such as your prototype building?
English is satisfactory. There are several typos and too complicated sentences in the document but nothing serious.
Reference list: reference 5 and 6 are almost identical – please delete number 6

Author Response
Dear authors,
Thank you very much for the interesting research. The topic of the manuscript is in accordance with the Journal requirements. I pretty much enjoyed reading it. The manuscript is very well structured and critical parts are explained. I appreciate the effort for writing it in a proper manner.
The dynamic behaviour of masonry buildings in the Mediterranean region is the topic researched among numerous researchers, and the OMA method is not very well accepted. Probably the reason is the price of fieldwork. I appreciate the work.
The authors thank the reviewer for the constructive and valuable comments. They attempted to improve the scientific quality of the manuscript by addressing all the below recommendations and remarks.
Here are my recommendations and remarks.
- Formating
The manuscript is not in the template. Introduction should be 1 and Description of the prototype building number 2. Figure 3 in o the wrong place and covers the title of Chapter 3
Corrected.
- Title
Maybe it can be more specific – maybe you can add …numerical simulations o a prototype building (or case study)
The title was modified to: “Operational modal analysis and non-linear dynamic simulations of a prototype low-rise masonry building“
- Abstract
ok
- Introduction
- Reference 1-4 – There is a lot more investigation on a full scale specimens, in example scholars from Slovenia, Croatia and Macedonia have many papers on the same topic. A lot of research was carried out in the IZIIS institute
The authors did not now about the IZIIS institute. Thank you for letting us know the research carried out in this institute. The references have been updated accordingly.
- Destructive tests give insight into the mechanical properties of a limited area where the tests are executed. Conversely, dynamic tests may deliver comprehensive knowledge into the homogenized mechanical parameters by updating a finite element model using experimental modal parameters. I partially agree. I think NDT or semi-NDT is a must. Mechanical properties are of a limited area but still better than pure guessing of the material properties. Also, the procedure by Borri et al. gives a visual classification of the masonry properties quite well. Maybe it can be beneficial for the paper to compare the visual classification and recommendations from Italian norms. The second part of the sentence: may deliver comprehensive knowledge of homogenized mechanical… - dynamic test can give an insight into a global behaviour of the structure, not in the mechanical properties. Of course, it gives the indication, but the differences can be more than 100%. Maybe these sentences should be modified.
The paragraph has been modified as follows: “Non Destructive Tests (NDT) or semi-NDT are necessary for masonry structures due to the intrinsic heterogeneity of their features and consequent variety of their mechanical properties.
Semi-NDT give insight into the mechanical properties of a limited area where the tests are executed.
Still, the mechanical properties of a limited area are more valuable than pure estimating based on visual inspections and classification approaches.
However, dynamic tests may deliver comprehensive knowledge into the homogenized mechanical parameters by updating a finite element model using experimental modal parameters. The model updating of a masonry structure using the modal parameters estimated from OMA can return information about the Elastic Modulus, weight and other parameters proper to address and refine structural investigations. Combined OMA and model updating is still an embryonal practice in masonry structures due to the high uncertainties related to estimating the modal parameters of low-rise masonry structures from OMA and the difficulty in matching the estimated response with the conventional, simplified structural models adopted for masonry structures.”
- OMA can return info about the elastic modulus – please explain how. I don’t understand how?
The authors corrected. Not directly OMA, but the combined use of OMA and model updating can deliver an estimate of the Young modulus.
- Reference 11 – why this one?
The reference was removed.
- Numerical analyses
- Why wasn’t the building modeled with the true geometry? It’s simplified but from my point of view at point Y1 (figure 7) the behaviour will not be as in the model because of the geometry change.
The angular inclination of Y1 in plan is negligible (< 15°): this consideration justifies the modelling of the masonry pier as a single pier, leading to an irrelevant variation of stiffness, allowing to reduce the burden of the EF model.
- Table 1 – again, it can be compared to a visual classification (MQI method by Borri et al.)
The Elastic Modulus of the masonries are evaluated by the approach called IQM, Indice di Qualità Muraria (Borri A., Castori, G., Corradi, M., De Maria, A., A method for the analysis and classification of historic masonry. Bulletin of Earthquake Engineering, 2014, 13, 2647-2665, 2015; Borri A. and De Maria, A., Scheda di valutazione dell’IQM. (allegato 3b.1-UR06-1). Rete dei Laboratori Universitarie di Ingegneria Sismica (RELUIS). Progetto esecutivo 2005-2008; Borri A. and De Maria, A., Qualità muraria secondo il metodo IQM: aggiornamento alla circolare esplicativa n. 7/2019. Structural, 222, 2019) in order to confirm the values reported in Table 1.
The Masonry Quality index, IQM, estimates the existing masonry based on qualitatively and quantitatively characteristics: material, section dimensions, joints, number of leaves, type of predominant material, presence and effectiveness of connections between leaves, and mortar types. Their values for the masonries of the building are:
- Irregular layout with masonry units embedded with good mortar: IQM=3.00
- Reinforced with grout injections: IQM=4
- Clay bricks: IQM=7
According the equation of the literature:
E= 731.51*e^(0.1548*IQM)
The Elastic Modulus associated is:
- Irregular layout with masonry units embedded with good mortar: E=1112 MPa
- Reinforced with grout injections: E=1358 MPa
- Clay bricks: E=1112 MPa
- Table 2 -it should be in the appendix. σ0 is from SAP? Do you think this is a good or ie. Flat jack method should be applied to get the σ0? From our research it’s impossible to get correct σ0 from the model.
Table 2 has been moved to the Appendix. The authors observed, in their experience, a minor discrepancy between the sigma_0 from the model and the one estimated in situ. Furthermore, the authors estimated the values of σ0 analytically and then assigned to the frame of the EF model, in order to improve the accuracy of the evaluation of the ultimate moment of the rocking hinges and the ultimate strength of shear hinges.
- Figure 8 – Why was the setup like this? Although I understand the concept, I will always recommend to add one more measuring point like on the drawing. The change in geometry influence the behaviour. This would be for sure seen in the Figure 11 if it was applied as I sketched.
The authors have only ten accelerometers and chose not to merge multiple setups to avoid error propagation in multi-setups estimations, already affected by the poor Signal-to-Noise ratio. Therefore, they chose the setups using 10 accelerometers based on the following criteria. The first, with the accelerometers at the corners aim at estimating the global modes, while the ones the accelerometers in a row aim at grasping the sole dynamic of the floor.
Model updating
- Table 4 – ranges of considered values seems quite high, especially for the examined building. Correlate it with some experimental research where NDT was used.
The minimum values, reported in Table 4, referred to the masonries, are selected according to the Normative dispositions, see the table below. In particular, the minumum value of the elastic modulus of the masonry composed by irregular masonry units, Em, is equal to the 0.5*(1020+1440)=1230 MPa; the mechanical characteristics are corrected by a correction factor equal to 1.7 to account the effectiveness of the strengthening interventions on the mechanical characteristics of the masonry. The minimum value of the elastic modulus of the masonry composed by clay bricks, Em, is equal to the 0.5*(1200+1800)=1500 Mpa. These values of the elastic moduli provided are the starting values in the model updating process.
The maximum values are deliberately set equal high values, leading to a satisfying range, to find the minimum of the selected parameters in the model updating process accurately. These values are higher than ones of the range reported in the Normative Code. The comparison is carried out for the un-reinforced masonry due to the availability of the experimental results, by the values of the retrofitted masonry follow the same considerations. For example, the Elastic Modulus of the retrofitted masonry with grout injections is evaluated as the elastic modulus of un-retrofitted masonry corrected with a coefficient equal to 1.7:
|
Elastic modulus of Masonry reinforced with grout injections [MPa] |
|||
|
C8.5.II (Italian Normative Code) |
IQM approach |
Experimental results from flat jack test on a specimen of masonry wall in L’Aquila (like example) |
|
|
mean |
max |
1112*1.7=1890.4 |
2413.02*1.7=41102 |
|
1020*1.7=1734 |
1440*1.7=2448 |
||
The founded values of the elastic moduli E, according to the IQM and experimental results, are higher than the Normative. Therefore, the authors justify the discrepancy due to the dispersion value of the masonry material and thus they chose to set a wide range of values in the model updating.
Numerical investigation of the structural role of the horizontal structure in non-linear dynamic field
The actual floor of the building is made of steel joists and brick tiles. Can you show the photo? Is it some prefab floor? It looks like standard prefabricated floors used in southern Europe in the last century.
The authors added a picture of the floor, see Fig. 16 (d).
The photo has been added.
- Table 6: HS3, HS1, HS5 – I don’t understand why numbers 3,1 and 5 were used? Is it a typo?
Corrected. HS1, HS2, HS3 indicate the different typologies of floors.
- Table 7 – you don’t need to change it, but it could be beneficial to have earthquakes from ie. Albania or Greece where this type of buildings is typical.
The authors thank the reviewer for the valuable suggestions. They will consider also earthquakes from that area in future research analyses.
- Page 17: Specifically, the use of the HS3 floor determines a reduction of the story drift equal to approximately 100% in both directions. Maybe it does, but recent research suggests that replacing traditional wooden floors with rigid diaphragms, i.e., RC floors, can induce some unwanted consequences such as cracks on the edges of the two materials or, in the worst scenario, disintegration and collapse of the masonry walls (ie. https://doi.org/10.3390/su13116353 and a lot of Italian articles)
The authors added a sentence marked in red explaining the cons of using rigid diaphragms.
- Conclusion
- At this aim, an in-depth experimental campaign is carried out through two ambient vibration tests of a selected building selected.
- The comparison between HS1 and HS2 start manifesting if the floor free length is higher than 4 m. Is it common to have floor free length higher than 4 m in buildings such as your prototype building?
It is pretty uncommon to have the floor free length higher than 4 m. The authors selected this building because, due to the significant free length of the floor, it would have been more straightforward to estimate the floor dynamics from OMA.
English is satisfactory. There are several typos and too complicated sentences in the document but nothing serious.
Reference list: reference 5 and 6 are almost identical – please delete number 6
Corrected.
Reviewer 2 Report
This paper investigates the dynamic behavior of an archetype low-rise masonry building using both experimental and numerical methods. They first described their prototype building then developed two FE models of the building to model its local and global behaviors. Afterwards, the authors measured the actual dynamic parameters of the prototype building using a set of sensors and updated the developed FE models accordingly. They used the updated FE models to study the effect of different parameters on the seismic response of low-rise masonry buildings. They also used the updated FE models in an incremental dynamic analysis (IDA) to develop fragility curves for masonry buildings. In general, the paper describes the scope of the work and techniques used very well but it suffers from some major and minor problems. The followings are the reviewer comments divided into major and minor comments.
Major Comments:
- The abstract is not written well in its current form. The authors should arrange their ideas and present them in a more organized and concise manner. Right now, I can’t tell the innovation and contribution in the presented effort.
- In the sentence staring with “The evaluation of dynamic …”, I would suggest that the authors add some references for studies on the dynamic behavior of masonry buildings.
- In the sentence starting with “Within the experimental analysis ..”, I would suggest that the authors mention the applied element method (AEM) along with the finite element method (FEM) due to its powerful capabilities in modeling and validating the dynamic behavior of structures under extreme loadings such as earthquakes. Some examples for the authors are Domaneschi et al. (2019) and Sediek et al. (2021).
- The authors oversimplified the constitutive relationship of the plastic hinges. Please comment on this in the manuscript.
- Figure 12 is not informative. I would suggest the authors to change the way of reporting the results of the optimization problem they solved in the model updating section.
- In Figure 13, the contour values/colors are missing. Please adjust.
Minor Comments:
- In the title, some words are capitalized and others not. Please correct.
- In the abstract, I suggest that the authors separate the sentence starting with “The first experimental setup …” into two sentences with the connection “whereas” or “while” between them.
- In the abstract, the sentence starting with “A parametric assessment ..” is vague. Please clarify.
- Modify the location of Fig. 3
- Change the name of section 6 to “Conclusions”
References:
Domaneschi, M., Paolo, G., and Scutiero, G. (2019). “A simplified method to assess generation of seismic debris for masonry structures.” Engineering Structures, 186 (December 2018), 306–320.
Sediek, O.A., El-Tawil, S., and McCormick, J., (2021) “Seismic Debris Field for Collapsed RC Moment Resisting Frame Buildings”, Journal of Structural Engineering, 147 (5), 04021045 https://doi.org/10.1061/(ASCE)ST.1943-541X.0002985
Author Response
This paper investigates the dynamic behavior of an archetype low-rise masonry building using both experimental and numerical methods. They first described their prototype building then developed two FE models of the building to model its local and global behaviors. Afterwards, the authors measured the actual dynamic parameters of the prototype building using a set of sensors and updated the developed FE models accordingly. They used the updated FE models to study the effect of different parameters on the seismic response of low-rise masonry buildings. They also used the updated FE models in an incremental dynamic analysis (IDA) to develop fragility curves for masonry buildings. In general, the paper describes the scope of the work and techniques used very well but it suffers from some major and minor problems. The followings are the reviewer comments divided into major and minor comments.
The authors thank the reviewer for the constructive and valuable comments. They attempted to improve the scientific quality of the manuscript by addressing all the below major and minor comments.
Major Comments:
- The abstract is not written well in its current form. The authors should arrange their ideas and present them in a more organized and concise manner. Right now, I can’t tell the innovation and contribution in the presented effort.
The abstract has been re-written in a more concise and organized way.
In the sentence staring with “The evaluation of dynamic …”, I would suggest that the authors add some references for studies on the dynamic behavior of masonry buildings.
The authors added references to the dynamic behavior of masonry buildings.
In the sentence starting with “Within the experimental analysis ..”, I would suggest that the authors mention the applied element method (AEM) along with the finite element method (FEM) due to its powerful capabilities in modeling and validating the dynamic behavior of structures under extreme loadings such as earthquakes. Some examples for the authors are Domaneschi et al. (2019) and Sediek et al. (2021).
The authors modified the sentence as suggested.
- The authors oversimplified the constitutive relationship of the plastic hinges. Please comment on this in the manuscript.
The authors chose to develop a Finite Element (FE) model, using an equivalent frame model, widely used for seismic analysis. The Finite Element model, based on a simplified modelling approach, ensures good results for global analysis of masonry structures in the non-linear field allowing also to reduce the computational costs of nonlinear analyses. Furthermore, the reliability of this modelling approach, that descends from a so-called lumped plasticity approach, was investigated by several scholars, also in the case of historical structures [47–49].
The definition of the all-plastic hinge respects the suggestion of the literature (Pasticier et al.2008): a rigid-perfectly plastic behavior. Furthermore, the strength in terms of ultimate moment (for the rocking hinges) and shear force (for the shear hinges) respect the equation recommended by the Eurocode 8, EC8-1 and the Italian Design Code.
- Figure 12 is not informative. I would suggest the authors to change the way of reporting the results of the optimization problem they solved in the model updating section.
The histograms in Figure 12 plot the average response in terms of the objective function, Fobj average value, frequency Ffrequency and modal shape Fmodal shapes respectively, similarly to an other research activity of the authors (I. Capanna, A. Aloisio, F. Di Fabio, and M. Fragiacomo, “Sensitivity Assessment of the Seismic Response of a Masonry Palace via Non-Linear Static Analysis: A Case Study in L’Aquila (Italy),” Infrastructures, vol. 6, no. 1, p. 8, 2021) in order to better evidence the influence of the uncertainties of materials characteristics and their scatter.
Since the figure is not clear, the authors removed it from the manuscript.
In Figure 13, the contour values/colors are missing. Please adjust.
The figure has been replaced by the following.
Minor Comments:
- In the title, some words are capitalized and others not. Please correct.
Corrected.
- In the abstract, I suggest that the authors separate the sentence starting with “The first experimental setup …” into two sentences with the connection “whereas” or “while” between them.
Corrected.
- In the abstract, the sentence starting with “A parametric assessment ..” is vague. Please clarify.
The abstract was entirely re-written as suggested by the reviewer within the major comments.
- Modify the location of Fig. 3
Corrected.
- Change the name of section 6 to “Conclusions”
Corrected.
References:
Domaneschi, M., Paolo, G., and Scutiero, G. (2019). “A simplified method to assess generation of seismic debris for masonry structures.” Engineering Structures, 186 (December 2018), 306–320.
Sediek, O.A., El-Tawil, S., and McCormick, J., (2021) “Seismic Debris Field for Collapsed RC Moment Resisting Frame Buildings”, Journal of Structural Engineering, 147 (5), 04021045 https://doi.org/10.1061/(ASCE)ST.1943-541X.0002985
Reviewer 3 Report
Dear authors,
The present article deals with the application of OMA on masonry (low-rise) buildings. It encompasses experimental and numerical studies, which is a fair point. In general, there is a lot of good material to do a publication. However, the reviewer asks for a major revision of the paper to better stress the contribution of the work to the scientific literature. What do the authors bring to research? Actually, the case of study is simple. The experimental campaign with only a few accelerometers and the numerical analyses simply emphasises the importance of the floor stiffness, which is probably one of the only contributions of the work. There is no comparison with the existing damage patterns. This major global comment ask for a significant revision to remove what is useless and stress what is innovative, new and of interest for the scientific community
Another general comment concerns the way to present things and the proofreading of the submitted manuscript. They were many typos that, in the reviewer’s opinion, show too little proofread of the manuscript by the authors. For a reader, the work is tough to understand, and the link between each section is not easy to understand. Too many times, the authors describe what they did in a chronological way. The reader does not the history of the study; he needs to understand clearly the aim, the results and the flow of the article. For now, this is not clear when reading.
As a major (though simple to address) general comment, the reviewer strongly advises adding line numbers in every manuscript to help the review process. It makes the review much easier (both for the authors and the reviewers). Additionally, the authors are asked to be consistent within the document regarding American or British English. They should choose one and keep it for the whole manuscript (including the title). Please also harmonise between ‘ and ’
Besides, the reviewer also noted some major and minor issues that need to be addressed. Note that they are not sorted by importance level.
Please also number the introduction section.
Figure 3 is misplaced. Please correct.
Figure 4 needs to be improved. Axis, orientation should be added. For now, the reviewer does not understand how the model expands. A good idea would be to superpose it with the actual structure. It will make things much easier to understand.
Figure 6 is missing: pls correct the figure numbers.
The second sentence of section 3 is not grammatically correct. Please add a verb and a subject.
Section 3. It is not clear how the authors use the two different setups. The authors insist on these two setups, but their difference (in terms of results they give) should be stressed in Section 3. For instance, the reader does not understand from which set up (first, second or both) the results depicted in Figure 11 come. This is a major issue.
Increase Figure 12 quality. Figure 12 is not clearly introduced. The reader does not really understand what is plotted (bars and scatter) and what is varied. All section 5.1 needs to be rewritten. Please correct. This is a major issue.
In table 4, the ranges used for Es2 are missing. Why?
Section 5.2: the values used for the range for Es1 (5000-15000) are different from the ones displayed in Table 4. Please rectify.
In the second paragraph of Section 5.2 Es1 is repeated twice, while one should be Es2.
In section 5.2, which set up results are used to compare with the numerical dynamic response?
The found specific weight of 13kN/m3 is very low. Can the authors comment on it?
Table 5 heading is reverted between Numerical and experimental.
The reviewer is lost when the authors talk about the first mode that they fitted twice (Section 5.2 and 5.3). What are the final frequency and MAC values found? In Section 5.2, do the values considered for Em and Eb influence the results (8.83Hz and MAC = 0.84)? Respectively, does Es1 and Es2 influence Section 5.3 results, at least for the first mode?
The last paragraph of page 14 is not clear at all. Does these stiffness parameters affect the frequencies and mode shapes? Why were they not calibrated through OMA? In general, the aim of section 5 should be better stated. Reference numbers for HS1, 2 3 are not consistent within the text and table.
Two words “selected” in the second paragraph of the conclusions.
Please highlight the authors’ contributions better in the conclusions. For instance, the effect of the floor diaphragm stiffness on global masonry behaviour is not really new. What do the authors add to the existing literature?
Author Response
Dear authors,
The present article deals with the application of OMA on masonry (low-rise) buildings. It encompasses experimental and numerical studies, which is a fair point. In general, there is a lot of good material to do a publication. However, the reviewer asks for a major revision of the paper to better stress the contribution of the work to the scientific literature. What do the authors bring to research? Actually, the case of study is simple. The experimental campaign with only a few accelerometers and the numerical analyses simply emphasises the importance of the floor stiffness, which is probably one of the only contributions of the work. There is no comparison with the existing damage patterns. This major global comment ask for a significant revision to remove what is useless and stress what is innovative, new and of interest for the scientific community
Another general comment concerns the way to present things and the proofreading of the submitted manuscript. They were many typos that, in the reviewer’s opinion, show too little proofread of the manuscript by the authors. For a reader, the work is tough to understand, and the link between each section is not easy to understand. Too many times, the authors describe what they did in a chronological way. The reader does not the history of the study; he needs to understand clearly the aim, the results and the flow of the article. For now, this is not clear when reading.
The authors thank the reviewer for his careful revision. They attempted to improve the scientific quality of the manuscript by focusing on the most important contributions of this research. Additionally, they further improved the proofreading of the manuscript.
As a major (though simple to address) general comment, the reviewer strongly advises adding line numbers in every manuscript to help the review process. It makes the review much easier (both for the authors and the reviewers). Additionally, the authors are asked to be consistent within the document regarding American or British English. They should choose one and keep it for the whole manuscript (including the title). Please also harmonise between ‘ and ’
Besides, the reviewer also noted some major and minor issues that need to be addressed. Note that they are not sorted by importance level.
The line numbers have been addressed and the document is revised according to American English.
Please also number the introduction section.
Corrected.
Figure 3 is misplaced. Please correct.
Corrected.
Figure 4 needs to be improved. Axis, orientation should be added. For now, the reviewer does not understand how the model expands. A good idea would be to superpose it with the actual structure. It will make things much easier to understand.
The authors replaced the figure with the following with orientation axes..
Figure 6 is missing: pls correct the figure numbers.
Figure 6 is the following:
The second sentence of section 3 is not grammatically correct. Please add a verb and a subject.
The sentence was corrected as follows:
“Specifically, the authors developed finite element (FE) models implemented in the software package SAP2000”
The authors: subject
Developed: verb
Section 3. It is not clear how the authors use the two different setups. The authors insist on these two setups, but their difference (in terms of results they give) should be stressed in Section 3. For instance, the reader does not understand from which set up (first, second or both) the results depicted in Figure 11 come. This is a major issue.
The authors clarified where the results shown in figures and tables come from. They all refer to the first setup. The following sentence has been added to the manuscript. “The first setup is used to characterize the global dynamics of the structure, while the second is used to identify with high accuracy the mode shapes characterized by a prevalent deformation of the floor.”
Increase Figure 12 quality. Figure 12 is not clearly introduced. The reader does not really understand what is plotted (bars and scatter) and what is varied. All section 5.1 needs to be rewritten. Please correct. This is a major issue.
The figure is unclear, the authors removed the figure.
In table 4, the ranges used for Es2 are missing. Why?
Corrected. It is a typo, they are the same of Es1.
Section 5.2: the values used for the range for Es1 (5000-15000) are different from the ones displayed in Table 4. Please rectify.
Corrected.
In the second paragraph of Section 5.2 Es1 is repeated twice, while one should be Es2.
Corrected.
In section 5.2, which set up results are used to compare with the numerical dynamic response?
The following sentence has been added: “The modal parameters estimated from the second setup are used for the model updating of the floor model.”.
The found specific weight of 13kN/m3 is very low. Can the authors comment on it?
The following sentence has been added. “The estimated value of the specific weight is very low compared to the expected values for masonry, beyond 18kN/m3. This finding reveals that traditional masonry buildings exhibit a significant scatter of the mechanical properties of existing masonry. Specifically, traditional masonry can be characterized by high porosity, evidence of poor masonry quality despite the apparent good texture of the exposed faces.”
Table 5 heading is reverted between Numerical and experimental.
Corrected.
The reviewer is lost when the authors talk about the first mode that they fitted twice (Section 5.2 and 5.3). What are the final frequency and MAC values found? In Section 5.2, do the values considered for Em and Eb influence the results (8.83Hz and MAC = 0.84)? Respectively, does Es1 and Es2 influence Section 5.3 results, at least for the first mode?
The following sentence was added: “In conclusion, the frequencies and MAC values in Figs. 14-15 derive from the modal identification of the global measurement setup. The modal identification in Section 5.2 aimed at describing the local mode shape of the floor. The sensitivity analyses revealed that the floor mechanical parameters do not influence the global structural dynamics. Therefore, these parameters were not used in the model updating of the global modes.”
The last paragraph of page 14 is not clear at all. Does these stiffness parameters affect the frequencies and mode shapes? Why were they not calibrated through OMA? In general, the aim of section 5 should be better stated. Reference numbers for HS1, 2 3 are not consistent within the text and table.
The following sentence was added: “In the following section, the authors first carry out the sensitivity analyses to assess the influence of the mechanical parameters of both the horizontal and vertical structural elements on the global dynamics. The analyses revealed that the dynamic of the floor does not influence that of the global dynamics in the considered ranges of the parameters. Therefore, this evidence allowed a separate model updating of the local dynamics of the floor (Section 5.2) and the global dynamics of the entire structure (Section 5.3).”
Two words “selected” in the second paragraph of the conclusions.
Corrected.
Please highlight the authors’ contributions better in the conclusions. For instance, the effect of the floor diaphragm stiffness on global masonry behaviour is not really new. What do the authors add to the existing literature?
The authors re-wrote the conclusions to better identify the findings of this research.
Reviewer 4 Report
The submitted manuscript investigates the influence of deformable and rigid floor diaphragm through operations modal analysis (OMA) and updates the numerical models accordingly to better predict the seismic response of a residential unreinforced masonry building. The objectives and outcomes are well explained, and the results support the existing knowledge in the literature. The comments are shared below.
1) “… The model updating of a masonry structure using the modal parameters estimated from OMA can return information about the Elastic Modulus, weight and other parameters useful to address and refine structural investigations … ”
This statement does not address the primary motivation of OMA, which is performed to find modal features of the buildings (namely, mode shapes, modal frequencies, damping ratio, etc.). Please take out the or re-write this sentence, considering the suggestion.
2) The uncertainty in the material properties and the sensitivity of the computational models to input parameters are the main challenges in the seismic assessment of unreinforced masonry structures. The authors should also refer to the suggested most recent studies in the literature regarding this subject.
DOI: 10.1016/j.engstruct.2021.112095, DOI: 10.1016/j.engstruct.2021.112620; DOI: 10.1061/(asce)cf.1943-5509.0001494
3) Please explain in the manuscript the considered ultimate shear displacement and bending rotation are experiment-based limits (these maximum values include any safety factors).
4) In Table-1, is the shear strength () equal to ()? Please, clarify in the text.
5) In Table-2, the moment unit is not correct.
6) “Specifically, the authors modelled the 3d finite element model (EFM) of the timber and RC floors.” This statement is meaningless (EFM ?). It seems that the authors want to state that 3D shell elements were used in the computational model. Please, re-write the statement.
7) I believe, in Table 7, PGA should be given in terms of gravitational acceleration (e.g., 0.3g)
8) In Table 6, HS# are different from the ones given in the text. Did the authors change the thickness of the slabs (for different slab materials) to match the validated natural frequencies? If so, how much different is the thickness values for HS2 and HS3 compared to their real applications?
9) The results are given in Figure 18 for which earthquake motion?
10) In all numerical models with different materials for slabs, is it assumed that slab and the load-bearing walls are fully connected?
Author Response
The submitted manuscript investigates the influence of deformable and rigid floor diaphragm through operations modal analysis (OMA) and updates the numerical models accordingly to better predict the seismic response of a residential unreinforced masonry building. The objectives and outcomes are well explained, and the results support the existing knowledge in the literature. The comments are shared below.
- “… The model updating of a masonry structure using the modal parameters estimated from OMA can return information about the Elastic Modulus, weight and other parameters useful to address and refine structural investigations … ” This statement does not address the primary motivation of OMA, which is performed to find modal features of the buildings (namely, mode shapes, modal frequencies, damping ratio, etc.). Please take out the or re-write this sentence, considering the suggestion.
The statement has been re-written.
2) The uncertainty in the material properties and the sensitivity of the computational models to input parameters are the main challenges in the seismic assessment of unreinforced masonry structures. The authors should also refer to the suggested most recent studies in the literature regarding this subject. DOI:10.1016/j.engstruct.2021.112095, DOI:10.1016/j.engstruct.2021.112620; DOI:10.1061/(asce)cf.1943- 5509.0001494
The authors introduced the suggested references.
[17] J. Gooch, M. J. Masia, and M. G. Stewart, "Application of stochastic numerical analyses in the assessment of spatially variable unreinforced masonry walls subkected to in-plane shear loading.", Eng. Struct., 235(ST5): 112095, 2021.
[18] B. Pulatsu, S. Gonen, E. Erdogmus, P. B. Lourenco, J. Lemos, P. R. Prakash, "In-plane structural performance of dry-joint stone masonry Walls: a spatial and non-spatial stochastic discontinuum analysis.", Eng. Struct., 242: 112620, 2021.
[19] R. M. Oinam, R. Sahoo, "Using metallic dampers to improve seismic performance of soft-story RC frames: experimental and numerical study", Journal of Perforormance of constructed facilities, 33(1): 1-18.
2) Please explain in the manuscript the considered ultimate shear displacement and bending rotation are experiment-based limits (these maximum values include any safety factors).
These suggestions have been introduced in the manuscript. The authors specify that the mechanical parameters are assigned according to the normative dispositions with a confidence factor equal to 1.35, while the mechanical characteristics of the plastic hinges are evaluated analytically. In any case, not experimental tests on the masonries have been conducted to investigate their mechanical parameters.
4) In Table-1, is the shear strength () equal to ()? Please, clarify in the text.
fv0 indicates the design shear strength with no axial force. The authors change the label in the Table 1.
5) In Table-2, the moment unit is not correct.
Corrected.
6) “Specifically, the authors modelled the 3d finite element model (EFM) of the timber and RC floors.” This statement is meaningless (EFM ?). It seems that the authors want to state that 3D shell elements were used in the computational model. Please, re-write the statement.
The statement has been re-written in a more concise and organized way.
“Specifically, the authors modelled three numerical models different for the typologies of floors: (i) HS1 is the label of the numerical model of the investigated building; (ii) HS2 is the label of the numerical model with timber wooden floors; (iii) HS3 is the label of the numerical model with RC floors. The modelling approach of masonry panels is descripted in the chapter 3.
7) I believe, in Table 7, PGA should be given in terms of gravitational acceleration (e.g., 0.3g)
The authors thank the reviewer for the valuable suggestions. They change the unit of PGA.
8) In Table 6, HS# are different from the ones given in the text. Did the authors change the thickness of the slabs (for different slab materials) to match the validated natural frequencies? If so, how much different is the thickness values for HS2 and HS3 compared to their real applications?
Corrected. The authors assign a thickness of the slab for the case study to match the validated natural frequencies, using the second setup of the experimental campaign carried out by the authors. For HS2 and HS3, the unavailability of experimental results leads to assign values of reference, suggested by the literature.
9) The results are given in Figure 18 for which earthquake motion?
The results depicted in Figure 18 are derived from the seven earthquake motions.
10) In all numerical models with different materials for slabs, is it assumed that slab and the loadbearing walls are fully connected?
In all numerical models, the slabs and the load bearing walls are assumed as perfectly and full connected.
Round 2
Reviewer 2 Report
I don't have any further comments. The paper is ready to be published in its current form.
Author Response
The authors thank the reviewers for the positive comments.
The paper was modified accordingly.
Reviewer 3 Report
Please provide a version with line numbers
Author Response
The paper was modified accordingly. The authors uploaded a version with line numbers.
Reviewer 4 Report
The authors addressed all the concerns and questions raised by the reviewer.
Author Response
The authors thank the reviewer for the positive comments.
Round 3
Reviewer 3 Report
The reviewer greatly thanks the authors for their modifications that made the paper much clearer.
Though he is not convinced about the high scientific contribution of the paper, the paper can now be accepted for publication. There is only minor checks that have to be addressed before the final version (no need to do another round of review, though).
Figure 4:
Do the axis represented on the Figure correspond to the one on Figure 2? The reviewer assumes so.
In the legend, the reviewers suggests adding the mention to “numerical model of the floor system” to make clear that it represents only one floor, which was not clear at first reading. Moreover the drawn spring (in red) close to the axis should be clearer, with a legend ideally.
Line 541 & 544 “Behaviour”: please check carefully the consistency of English writing style.
Author Response
The paper was modified according to the reviewer suggestions.